# QuACK: Accelerating Gradient-Based Quantum Optimization with Koopman Operator Learning

**Di Luo** *
Center for Theoretical Physics,
Massachusetts Institute of Technology,
Cambridge, MA 02139, USA
Department of Physics, Harvard University,
Cambridge, MA 02138, USA
The NSF AI Institute for Artificial
Intelligence and Fundamental Interactions
diluo@mit.edu

**Jiayu Shen** *
Department of Physics,
University of Illinois, Urbana-Champaign
Urbana, IL 61801, USA
Illinois Quantum Information Science
and Technology Center
Illinois Center for Advanced Studies
of the Universe
jiayus3@illinois.edu

**Rumen Dangovski**
Department of Electrical Engineering
and Computer Science,
Massachusetts Institute of Technology
Cambridge, MA 02139, USA
rumenrd@mit.edu

**Marin Soljačić**
Department of Physics,
Massachusetts Institute of Technology
Cambridge, MA 02139, USA
soljacic@mit.edu

## Abstract

Quantum optimization, a key application of quantum computing, has traditionally been stymied by the linearly increasing complexity of gradient calculations with an increasing number of parameters. This work bridges the gap between Koopman operator theory, which has found utility in applications because it allows for a linear representation of nonlinear dynamical systems, and natural gradient methods in quantum optimization, leading to a significant acceleration of gradient-based quantum optimization. We present Quantum-circuit Alternating Controlled Koopman learning (QuACK), a novel framework that leverages an alternating algorithm for efficient prediction of gradient dynamics on quantum computers. We demonstrate QuACK's remarkable ability to accelerate gradient-based optimization across a range of applications in quantum optimization and machine learning. In fact, our empirical studies, spanning quantum chemistry, quantum condensed matter, quantum machine learning, and noisy environments, have shown accelerations of more than 200x speedup in the overparameterized regime, 10x speedup in the smooth regime, and 3x speedup in the non-smooth regime. With QuACK, we offer a robust advancement that harnesses the advantage of gradient-based quantum optimization for practical benefits.

## 1   Introduction

The dawn of quantum computing has ushered in a new era of technological advancement, presenting a paradigm shift in how we approach complex computational problems. Central to these endeavors are Variational Quantum Algorithms (VQAs) [14], which are indispensable for configuring and optimizing quantum computers. These algorithms serve as the backbone of significant domains,

---

*Co-first authors

37th Conference on Neural Information Processing Systems (NeurIPS 2023).

including quantum optimization [54] and quantum machine learning (QML) [8]. VQAs have also influenced advances in various other quantum learning applications [17, 24, 32, 45, 28, 43, 27, 67]

At the core of VQAs lies the challenge of effectively traversing and optimizing the high-dimensional parameter landscapes that define quantum systems. To solve this, there are two primary strategies: gradient-free [72] and *gradient-based* methods [38]. Gradient-free methods, while straightforward and widely used in practice, do not provide any guarantees of convergence, which often results in suboptimal solutions. On the other hand, gradient-based methods, such as those employed by the Variational Quantum Eigensolver (VQE) [59, 78], *offer guarantees* for convergence. This characteristic has facilitated their application across a multitude of fields, such as high-energy physics [34, 65], condensed matter physics [83], quantum chemistry [58], and important optimization problems such as max-cut problem [19, 23].

More specifically, recent developments in overparameterization theory show that gradient-based methods like gradient descent provide convergence guarantees in a variety of quantum optimization tasks [37, 44, 87]. New research indicates these methods surpass gradient-free techniques such as SPSA [72] in the context of differentiable analog quantum computers [38]. Notably, the quantum natural gradient, linked theoretically with imaginary time evolution, captures crucial geometric information, thereby enhancing quantum optimization [73].

Despite these advantages, the adoption of gradient-based methods in quantum systems is not without its obstacles. The computation of gradients in these hybrid quantum-classical systems is notoriously *resource-intensive* and scales linearly with the number of parameters. This computational burden presents a significant hurdle, limiting the practical application of these methods on quantum computers, and thus, the full potential of quantum optimization and machine learning. Hence, we pose the question, "*Can we accelerate gradient-based quantum optimization?*" This is crucial to harness the theoretical advantages of convergence for practical quantum optimization tasks. In this work, we answer our question affirmatively by providing order-of-magnitude speedups to VQAs. Namely, our contributions are as follows:

- By perceiving the optimization trajectory in quantum optimization as a dynamical system, similarly to [15, 61] we bridge quantum natural gradient theory, overparameterization theory, and Koopman operator learning theory [36], which allows for a linear represenation of nonlinear dynamical systems and thus is useful in applications.

- We propose Quantum-circuit Alternating Controlled Koopman Operator Learning (QuACK), a new algorithm grounded on this theory. We scrutinize its spectral stability, convex problem convergence, complexity of speedup, and non-linear extensions using sliding window and neural network methodologies.

- Through extensive experimentation in fields like quantum many-body physics, quantum chemistry, and quantum machine learning, we underscore QuACK's superior performance, achieving speedups over 200x, 10x, 3x, and 2x–5.5x in overparameterized, smooth, non-smooth and noisy regimes respectively.

## 2 Related Work

**Quantum Optimization Methods.** Owing to prevailing experimental constraints, quantum optimization has frequently employed gradient-free methods such as SPSA, COBYLA, and Bayesian optimization [72, 70, 76], among others that alleviate the challenges inherent in quantum optimization [89, 25, 86]. Regarding gradient-based methods, the quantum natural gradient [73] exhibits compelling geometric properties, and conventional gradient methods such as SGD and Adam are applicable as well. Recent developments in overparameterization theory [37, 44, 87] have provided assurances for SGD's convergence in quantum optimization. We demonstrate an example where the gradient-based methods find the minimum while the gradient-free method gets trapped in Appendix C. While meta-learning techniques [81, 33, 85] have been explored to accelerate optimization across various tasks, our focus in this work is on the acceleration of gradient-based quantum optimization, given the appealing theoretical properties of such methods. We emphasize, however, that our approach is complementary to meta-learning and could be integrated as a subroutine for potential enhancements.

Some additional lines of work are related to our study. You et al. [88] analyze the convergence of quantum neural networks through the lens of the neural tangent kernel. Similarly, working through

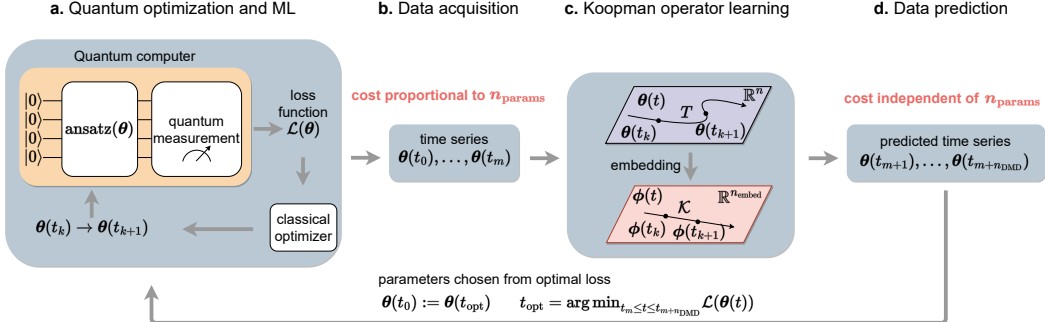

Figure 1: QuACK: Quantum-circuit Alternating Controlled Koopman Operator Learning. (a) Parameterized quantum circuits process information; loss function is evaluated via quantum measurements. Parameter updates for the quantum circuit are computed by a classical optimizer. (b) Optimization history forms a time series, the computational cost of which is proportional to the number of parameters. (c) Koopman operator learning finds an embedding of data with approximately linear dynamics from time series in (b). (d) Koopman operator predicts parameter updates with computational cost independent of the number of parameters. Loss from predicted parameters is evaluated, and optimal parameters are used as starting point for the next iteration.

an effective quantum neural tangent kernel theory, Wang et al. [82] propose symmetric pruning to improve the loss landscape and the convergence of quantum neural networks. Finally, García-Martín et al. [20] study the effect of noise on overparameterization in quantum neural networks.

**Koopman Theory.** The Koopman operator theory, originating from the 1930s, furnishes a framework to understand dynamical systems [36, 80]. The theory has evolved, incorporating tools such as Dynamic Mode Decomposition (DMD) [68] (connected to Koopman mode decomposition [50, 66]) extended-DMD [84, 15, 61, 10, 3, 30, 79, 9], and machine learning approaches [48, 41, 4, 64]. While there have been successful applications in neural network optimization [16, 77, 60] and quantum mechanics [22, 35], the linking of this theory with quantum natural gradient and overparameterization theory for optimization acceleration, as we have done, is novel to our knowledge. Ours and the above contributions are built upon works on machine learning using Koopman operator theory [52, 53, 55, 71, 42].

## 3    Background

**Variational Quantum Algorithm (VQA).**    For a quantum mechanical system with $N$ qubits, the key object that contains all the information of the system is called a wave function $\psi$. It is an $l_2$-normalized complex-valued vector in the $2^N$-dimensional Hilbert space. A parameterized quantum circuit encodes a wave function as $\psi_{\boldsymbol{\theta}}$ using a set of parameters $\boldsymbol{\theta} \in \mathbb{R}^{n_{\text{params}}}$ via an ansatz layer on a quantum circuit in a quantum computer, as shown in the top-left part of Figure 1. The number of parameters, $n_{\text{params}}$, is typically chosen to be polynomial in the number of qubits $N$, which scales much slower than the $2^N$-scaling of the dimension of $\psi$ itself in the original Hilbert space.

Variational quantum eigensolver (VQE) aims to solve minimal energy wave function of a Hamiltonian with parameterized quantum circuit. A Hamiltonian $\mathcal{H}$ describes interactions in a physical system, which mathematically is a Hermitian operator acting on the wave functions. The energy of a wave function is given by $\mathcal{L}(\psi) = \langle \psi | \mathcal{H}\psi \rangle$. VQE utilizes this as a loss function to find the optimal $\boldsymbol{\theta}^* = \arg\min_{\boldsymbol{\theta}} \mathcal{L}(\boldsymbol{\theta}) = \arg\min_{\boldsymbol{\theta}} \langle \psi_{\boldsymbol{\theta}} | \mathcal{H}\psi_{\boldsymbol{\theta}} \rangle$. Quantum machine learning follows a similar setup, aiming to minimize $\mathcal{L}(\boldsymbol{\theta})$ involved data with parameterized quantum circuits $\psi_{\boldsymbol{\theta}}$, which in QML are usually referred to as quantum neural networks.

To minimize the loss function, one can employ a gradient-based classical optimizer such as Adam, which requires calculating the gradient $\partial \mathcal{L}/\partial \theta_i$. In the quantum case, one usually has to explicitly evaluate the loss with a perturbation in each direction $i$, for example, using the parameter-shift rule [51, 69]: $(\mathcal{L}(\theta_i + \pi/2) - \mathcal{L}(\theta_i - \pi/2))/2$. This leads to a linear scaling of $n_{\text{params}}$ computational cost, making quantum optimization significantly more expensive than classical backpropagation

which computational complexity is independent of $n_{\text{params}}$, while the required memory is still proportional to $n_{\text{params}}$. It is worth noting that the classical computational components involved in VQE, even including training neural-network-based algorithms in the following sections, typically are much cheaper than the quantum gradient cost given the scarcity of quantum resources in practice.

**Quantum Natural Gradient.** The quantum natural gradient method [73] generalizes the classical natural gradient method in classical machine learning [1] by extending the concept of probability to complex-valued wave functions. It is also theoretically connected to imaginary time evolution [73]. In the context of parameter optimization, the natural gradient for updating the parameter $\theta$ is governed by a nonlinear differential equation $\frac{d}{dt}\boldsymbol{\theta}(t) = -\eta F^{-1}\nabla_{\boldsymbol{\theta}}\mathcal{L}(\boldsymbol{\theta}(t))$, where $\eta$ denotes the scalar learning rate, and $F$ represents the quantum Fisher Information matrix defined as $F_{ij} = \langle\partial\psi_{\boldsymbol{\theta}}/\partial\theta_i|\partial\psi_{\boldsymbol{\theta}}/\partial\theta_j\rangle - \langle\partial\psi_{\boldsymbol{\theta}}/\partial\theta_i|\psi_{\boldsymbol{\theta}}\rangle\langle\psi_{\boldsymbol{\theta}}|\partial\psi_{\boldsymbol{\theta}}/\partial\theta_j\rangle$.

**Koopman Operator Learning.** Consider a dynamical system characterized by a set of state variables $x(t) \in \mathbb{R}^n$, governed by a transition function $T$ such that $x(t + 1) = T(x(t))$. According to the Koopman operator theory articulated by Koopman, a linear operator $\mathcal{K}$ and a function $g$ exist, satisfying $\mathcal{K}g(x(t)) = g(T(x(t))) = g(x(t + 1))$, where $\mathcal{K}$ represents the Koopman operator. Generally, this operator can function in an infinite-dimensional space. However, when $\mathcal{K}$ is restricted to a finite dimensional invariant subspace with $g : \mathbb{R}^n \to \mathbb{R}^m$, the Koopman operator can be depicted as a Koopman matrix $K \in \mathbb{R}^{m \times m}$. Data acquired from the dynamics are needed to compute the Koopman operator [12, 11]. The standard Dynamic Mode Decomposition (DMD) approach assumes $g$ to be the identity function, predicated on the notion that the underlying dynamics of $x$ are approximately linear, *i.e.*, $T$ operates as a linear function. The extended-DMD method broadens this scope, utilizing additional feature functions such as polynomial and trigonometric functions as the basis functions for $g$. Further enhancing this approach, machine learning methods for the Koopman operator leverage neural networks as universal approximators for learning $g$ [48].

# 4 QuACK - Quantum-circuit Alternating Controlled Koopman Operator Learning

**QuACK.** Our QuACK algorithm, illustrated in Algorithm 1 and Figure 1, employs a quantum circuit $\psi_{\boldsymbol{\theta}}$ with parameters $\boldsymbol{\theta}$ for quantum optimization or QML tasks. In Panel (a) of Figure 1 the loss function $\mathcal{L}(\cdot)$ is stochastically evaluated through quantum measurements, and a classical optimizer updates the parameters. In Panel (b) following $m$ gradient optimization steps, we obtain a time series of parameter updates $\boldsymbol{\theta}(t_0), \boldsymbol{\theta}(t_1), \ldots, \boldsymbol{\theta}(t_m)$. [2] Then in Panel (c) this series is utilized by the Koopman operator learning algorithm to find an embedding for approximately linear dynamics. In Panel (d) this approach predicts the parameter updates for $n_{\text{DMD}}$ future gradient dynamics steps and calculates $\boldsymbol{\theta}(t_{m+1}), \boldsymbol{\theta}(t_{m+2}), \ldots, \boldsymbol{\theta}(t_{m+n_{\text{DMD}}})$.

In each time step, the parameters are set directly in the quantum circuit for loss function evaluation via quantum measurements. This procedure has constant cost in terms of the number of parameters, same as the forward evaluation cost of the loss function. From the $n_{\text{DMD}}$ loss function values, we identify the lowest loss and the corresponding optimal time

---

**Algorithm 1** QuACK

**Input:** Quantum circuit $\psi_{\boldsymbol{\theta}}$, Loss function $\mathcal{L}(\cdot)$, Iterations $n_{\text{iter}}$, Koopman operator learning parameters $m$ (num. simulation. steps $n_{\text{sim}}$), $n_{\text{DMD}}$ (num. DMD steps).
**Output:** Optimal parameters $\boldsymbol{\theta}^*$
**Initialize:** $\boldsymbol{\theta}(t_0)$ randomly
**for** $i = 0$ to $n_{\text{iter}} - 1$ **do**
    *Quantum optimization or QML:*
    **for** $k = 0$ to $m - 1$ **do**
        Compute gradient $\nabla_{\boldsymbol{\theta}}\mathcal{L}(\boldsymbol{\theta}(t_k))$ and update $\boldsymbol{\theta}(t_k) \to \boldsymbol{\theta}(t_{k+1})$ using classical gradient-based optimizer
        Compute $\mathcal{L}(\boldsymbol{\theta}(t_{k+1}))$
    **end for**
    *Koopman operator learning:*
    Train Koopman operator $\mathcal{K}$ and predict optimization trajectory
    *Optimal parameters selection:*
    **for** $k = m + 1$ to $m + n_{\text{DMD}}$ **do**
        Compute $\mathcal{L}(\boldsymbol{\theta}(t_k))$
    **end for**
    Determine $t_{\text{opt}}$ and set $\boldsymbol{\theta}(t_0) \leftarrow \boldsymbol{\theta}(t_{\text{opt}})$
**end for**

---

[2] In this work we will interchangibly use $m$ and $n_{\text{sim}}$ to denote the same hyperparameter. For ease of notation, we use $m$ when indexing or looping, and $n_{\text{sim}}$ when we control for that parameter and plot dependencies.

$t_{\text{opt}} = \arg\min_{t_m \leq t \leq t_{m+n_{\text{DMD}}}} \mathcal{L}(\boldsymbol{\theta}(t))$. This step includes the last VQE iteration $t_m$ to prevent degradation in case of inaccurate DMD predictions. The algorithm iteratively alternates the simulation steps with the Koopman steps for $n_{\text{iter}}$ steps, similarly to [77]. To facilitate the Koopman operator learning algorithm, we introduce DMD, Sliding Window DMD and neural DMD into QuACK as follows.

**DMD and Sliding Window DMD.** Dynamic Mode Decomposition (DMD) employs a linear fit for vector dynamics $\boldsymbol{\theta} \in \mathbb{R}^n$, where $\boldsymbol{\theta}(t_{k+1}) = K\boldsymbol{\theta}(t_k)$. By concatenating $\boldsymbol{\theta}$ at $m+1$ consecutive times, we define $\boldsymbol{\Theta}(t_k) := [\boldsymbol{\theta}(t_k)\,\boldsymbol{\theta}(t_{k+1})\cdots\boldsymbol{\theta}(t_{k+m})]$. We create data matrices $\boldsymbol{\Theta}(t_0)$ and $\boldsymbol{\Theta}(t_1)$, the latter being the one-step evolution of the former. In approximately linear dynamics, $K$ is constant for all $t_k$, and hence $\boldsymbol{\Theta}(t_1) \approx K\boldsymbol{\Theta}(t_0)$. The best fit occurs at the Frobenius loss minimum, given by $K = \boldsymbol{\Theta}(t_1)\boldsymbol{\Theta}(t_0)^+$, where $+$ is the Moore-Penrose inverse.

When the dynamics of $\boldsymbol{\theta}$ is not linear, we can instead consider a time-delay embedding with a sliding window and concatenate the steps to form an extended data matrix [18]

$$\boldsymbol{\Phi}(\boldsymbol{\Theta}(t_0)) = [\boldsymbol{\phi}(t_0) \quad \boldsymbol{\phi}(t_1) \quad \cdots \quad \boldsymbol{\phi}(t_m)] = \begin{bmatrix} \boldsymbol{\theta}(t_0) & \boldsymbol{\theta}(t_1) & \cdots & \boldsymbol{\theta}(t_m) \\ \boldsymbol{\theta}(t_1) & \boldsymbol{\theta}(t_2) & \cdots & \boldsymbol{\theta}(t_{m+1}) \\ \vdots & \vdots & \ddots & \vdots \\ \boldsymbol{\theta}(t_d) & \boldsymbol{\theta}(t_{d+1}) & \cdots & \boldsymbol{\theta}(t_{m+d}) \end{bmatrix}. \quad (1)$$

$\boldsymbol{\Phi}$ is generated by a sliding window of size $d+1$ at $m+1$ consecutive time steps. Each column of $\boldsymbol{\Phi}$ is a time-delay embedding for $\boldsymbol{\Theta}$, and the different columns $\boldsymbol{\phi}$ in $\boldsymbol{\Phi}$ are embeddings at different starting times. The time-delay embedding captures some nonlinearity in the dynamics of $\boldsymbol{\theta}$, with $\boldsymbol{\Theta}(t_{d+1}) \approx K\boldsymbol{\Phi}(\boldsymbol{\Theta}(t_0))$, where $K \in \mathbb{R}^{n \times n(d+1)}$. The best fit is given by

$$K = \arg\min_K \|\boldsymbol{\Theta}(t_{d+1}) - K\boldsymbol{\Phi}(\boldsymbol{\Theta}(t_0))\|_F = \boldsymbol{\Theta}(t_{d+1})\boldsymbol{\Phi}(\boldsymbol{\Theta}(t_0))^+. \quad (2)$$

During prediction, we start with $\boldsymbol{\theta}(t_{m+d+2}) = K\boldsymbol{\phi}(t_{m+1})$ and update from $\boldsymbol{\phi}(t_{m+1})$ to $\boldsymbol{\phi}(t_{m+2})$ by removing the oldest data and adding new predicted data. This iterative prediction is performed via $\boldsymbol{\theta}(t_{k+d+1}) = K\boldsymbol{\phi}(t_k)$. Unlike the approach of Dylewsky et al. [18], we do not use an additional SVD before DMD, and our matrix $K$ is non-square. We term this method Sliding Window DMD (SW-DMD), with standard DMD being a specific case when the sliding window size is 1 ($d = 0$). The time-delay embedding is related Takens' theorem [75] with similar implementations in Hankel DMD [3] and streaming DMD [26, 21].

**Neural DMD.** To provide a better approximation to the nonlinear dynamics, we ask whether the hard-coded sliding window transformation $\boldsymbol{\Phi}$ can be a neural network. Thus, by simply reformulating $\boldsymbol{\Phi}$ in Eq. 2 as a neural network, we formulate a natural neural objective for Koopman operator learning as $\arg\min_{K,\alpha} \|\boldsymbol{\Theta}(t_{d+1}) - K\boldsymbol{\Phi}_\alpha(\boldsymbol{\Theta}(t_0))\|_F$,, where $K \in \mathbb{R}^{N_{in} \times N_{out}}$ is a linear Koopman operator and $\boldsymbol{\Phi}_\alpha(\boldsymbol{\Theta}(t_0))$ is a nonlinear neural embedding by a neural network $\boldsymbol{\Phi}_\alpha$ with parameters $\alpha$. $\boldsymbol{\Phi}_\alpha := \text{NN}_\alpha \circ \boldsymbol{\Phi}$ is a composition of the neural network architecture $\text{NN}_\alpha$ and the sliding window embedding $\boldsymbol{\Phi}$ from the previous section.

Drawing from the advancements in machine learning for DMD, we introduce three methods: MLP-DMD, CNN-DMD, and MLP-SW-DMD. MLP-DMD uses a straightforward MLP architecture for $\boldsymbol{\Phi}$, comprising two linear layers with an ELU activation and a residual connection, as shown in the Appendix. Unlike MLP-DMD, CNN-DMD incorporates information between simulation steps, treating these steps as the temporal dimension and parameters as the channel dimension of a 1D CNN encoder shown in the Appendix. To avoid look-ahead bias, we use causal masking of the CNN kernels, and the architecture includes two 1D-CNN layers with a bottleneck middle channel number of 1 and ELU activation to prevent overfitting. MLP-SW-DMD is similar to MLP-DMD but includes the time-delay embedding in the input, a feature we also add to CNN-DMD. As a result, MLP-DMD and CNN-DMD extend the principles of DMD, whereas MLP-SW-DMD and CNN-DMD with time-delay embedding generalize from SW-DMD. More details are available in the Appendix.

**Limitations.** QuACK is currently only equipped with relatively simple neural network architecture while more advanced architectures like Transformer can be explored for future work. Even though QuACK reduces the number of necessary gradient steps largely and thus achieves speedup, training

of VQA with a few gradient steps is still required to obtain the data for Koopman operator learning. $n_{\mathrm{sim}}$ for the training gradient steps in our design is a hyperparameter (see ablation study in Appendix), and we do not have a theoretical formula for it yet.

## 5 Theoretical Results

**Connection to Quantum Nature Gradient.** While there exists a Koopman embedding that can linearize a given nonlinear dynamics, we provide more insights between Koopman theory and gradient-based dynamics under quantum optimization. The dynamical equation from quantum natural gradients $d\boldsymbol{\theta}(t)/dt = -\eta F^{-1}\nabla_{\boldsymbol{\theta}}\mathcal{L}(\boldsymbol{\theta}(t))$ is equivalent to $d\psi_{\boldsymbol{\theta}}(t)/dt = -\mathbb{P}_{\psi_{\boldsymbol{\theta}}}\mathcal{H}\psi_{\boldsymbol{\theta}}(t)$, where $F$ is the quantum Fisher information matrix, and $\mathbb{P}_{\psi_{\boldsymbol{\theta}}}$ represents a projector onto the manifold of the parameterized quantum circuit. When the parameterized quantum circuit possesses full expressivity, it covers the entire Hilbert space, resulting in $d\psi_{\boldsymbol{\theta}}(t)/dt = -\mathcal{H}\psi_{\boldsymbol{\theta}}(t)$. This is a linear differential equation in the vector space of unnormalized wave functions. The normalized $\psi_{\boldsymbol{\theta}}$ from the parameterized quantum circuit together with additional normalization factor function $N(\boldsymbol{\theta})$ can serve as an embedding to linearize the quantum natural gradient dynamics. For a relative short time scale, the normalized $\psi_{\boldsymbol{\theta}}$ already provides a good linearization which only differs from the exact dynamics by the normalization change in the short time scale. The special underlying structure of quantum natural gradient may make it easier to learn the approximate linear dynamics for Koopman operator learning.

**Connection to Overparameterization Theory.** Recent advancement on overparameterization theory [37, 44, 87] finds that when the number of parameters in the quantum circuit exceed a certain limit, linear convergence of the variational quantum optimization under gradient descent is guaranteed. You et al. [87] shows that the normalized $\psi_{\boldsymbol{\theta}}$ from the parameterized quantum circuit in the overparatermization regime follows a dynamical equation which has a dominant part linear in $\psi_{\boldsymbol{\theta}}$ with additional perturbation terms. Similar to the quantum natural gradient, the overparameterization regime could provide an effective structure to simplify the Koopman operator learning with approximate linear dynamics.

### 5.1 Stability Analysis

We first note that the direct application of DMD without alternating the controlled scheme will lead to unstable or trivial performance in the asymptotic limit.

**Theorem 5.1.** *Asymptotic DMD prediction for quantum optimization is trivial or unstable.*

*Proof.* The asymptotic dynamics from DMD prediction is given by $\boldsymbol{\theta}(T) = K^T \boldsymbol{\theta}(t_0)$ for $T \to \infty$. It follows that $\boldsymbol{\theta}(T) \to w_m^T v_m$, where $w_m$ and $v_m$ are the largest magnitude eigenvalue and the corresponding eigenvector of the Koopman operator $K$. If $|w_m| < 1$, then $\boldsymbol{\theta}(T)$ will converge to zero, and the quantum circuit will reach a trivial fixed state. If $|w_m| \geq 1$, $\boldsymbol{\theta}(T)$ will keep oscillating or growing unbounded. For a unitary gate $U(\theta) = e^{-i\theta P}$ where $P$ is a Pauli string, $U(\theta)$ is periodic with respect to $\theta$. The oscillation or unbounded growing behavior of $\boldsymbol{\theta}(T)$ will lead to oscillation of the unitary gate in the asymptotic limit resulting in unstable performance. $\square$

The above issue is also found to exist in numerical results plotted in Appendix, indicating that the eigenvalues of the Koopman operators from quantum optimization dynamics can lead to trivial or unstable DMD prediction, which motivates us to develop QuACK. Indeed, our QuACK has controllable performances given by the following

**Theorem 5.2.** *In each iteration of QuACK, the optimal parameters $\boldsymbol{\theta}(t_{\mathrm{opt}})$ yield an equivalent or lower loss than the $m$-step gradient-based optimizer.*

*Proof.* From $t_{\mathrm{opt}} = \arg\min_{t_m \leq t \leq t_m + n_{\mathrm{DMD}}} \mathcal{L}(\boldsymbol{\theta}(t))$, we have $\mathcal{L}(\boldsymbol{\theta}(t_{\mathrm{opt}})) \leq \mathcal{L}(\boldsymbol{\theta}(t_m))$ where $\mathcal{L}(\boldsymbol{\theta}(t_m))$ is the final loss from the $m$-step gradient-based optimizer. $\square$

As long as QuACK is able to capture certain dynamical modes that decrease the loss, then it will produce a lower loss even when the predicted dynamics does not exactly follow the same trend of the $m$-step gradient descent updates. We also note that it is possible for QuACK to converge to a different

local minimum than the baseline gradient-based optimizer, but our experiments generally demonstrate that our QuACK achieves accuracy the same as or even better than the baseline with much faster convergence. Our procedure is robust against long-time prediction error and noise with much fewer gradient calculations, which can efficiently accelerate the gradient-based methods. Furthermore, our scheme has the following important *implication*, the proof of which we provide in Appendix.

**Corollary 5.3.** *QuACK achieves superior performance over the asymptotic DMD prediction.*

## 5.2 Complexity and Speedup Analysis

To achieve a certain target loss $\mathcal{L}_{\text{target}}$ near convergence, the traditional gradient-based optimizer as a baseline takes $T_b$ gradient steps. To achieve the same $\mathcal{L}_{\text{target}}$, QuACK takes $T_{Q,t} = T_{Q,1} + T_{Q,2}$ steps where $T_{Q,1}$ ($T_{Q,2}$) are the total numbers of QuACK training (prediction) steps. We denote the ratio between the computational costs of the baseline and QuACK as $s$, which serves as the speedup ratio from QuACK. Our definition of speedup has some difference with the speedup defined in Ref. [16] although shares a similar spirit. In gradient-based optimization, the computational costs of each step of QuACK training and prediction are different, with their ratio defined as $f(p)$ where $p := n_{\text{params}}$. In general, $f(p)$ is $\Omega(p)$ since only QuACK training, not prediction, involves gradient steps.

**Theorem 5.4.** *With a baseline gradient-method, the speedup ratio $s$ is in the following range*

$$a \leq s \leq a \frac{f(p)(n_{\text{sim}} + n_{\text{DMD}})}{f(p)n_{\text{sim}} + n_{\text{DMD}}}. \tag{3}$$

*where $a := T_b/T_{Q,t}$. In the limit of $n_{\text{iter}} \to \infty$ the upper bound can be achieved.*

We present the proof in the Appendix. $a$ is a metric of the accuracy of Koopman prediction. Higher $a$ means better prediction, and a perfect prediction has $a = 1$. The exact form of $f(p)$ depends on the details of the gradient method. For example, we have $f(p) = 2p + 1$ for parameter-shift rules and $f(p) \sim p^2 + p$ for quantum natural gradients [49]. If $a$ is fixed, then the upper bound of $s$ in Eq. 13 increases when $p$ increases with an asymptote $a(n_{\text{sim}} + n_{\text{DMD}})/n_{\text{sim}}$ at $p \to \infty$. The upper bound in Eq. 13 implies $s \leq af(p)$ where the equal sign is achieved in the limit $n_{\text{DMD}}/n_{\text{sim}} \to \infty$. If $a = 1$ one could in theory achieve $f(p)$-speedup, at least linear in $n_{\text{params}}$.

In practice, the variable $a$ is influenced by $n_{\text{sim}}$ and $n_{\text{DMD}}$. A decrease in $a$ can occur when $n_{\text{sim}}$ diminishes or $n_{\text{DMD}}$ enlarges, as prediction accuracy may drop if the optimization dynamics length for QuACK training is insufficient or the prediction length is too long. Moreover, estimating the gradient for each parameter requires a specific number of shots $n_{\text{shots}}$ on a fixed quantum circuit. Quantum measurement precision usually follows the standard quantum limit $\sim 1/\sqrt{n_{\text{shots}}}$, hence a finite $n_{\text{shots}}$ may result in noisy gradients for Koopman operator learning, influencing $a$. Intriguingly, when the prediction aligns with the pure VQA, QuACK's loss might decrease faster than pure baseline VQA, leading to $a > 1$ and a higher speedup. This could be due to dominant DMD spectrum modes with large eigenvalues aligning with the direction of fast convergence in the $\boldsymbol{\theta}$-space.

# 6 Experiments

## 6.1 Experimental Setup

We adopt the *Relative loss* metric for benchmark, which is $(\mathcal{L} - \mathcal{L}_{\text{min, full VQA}})/(\mathcal{L}_{\text{initial, full VQA}} - \mathcal{L}_{\text{min, full VQA}})$, where $\mathcal{L}$ is the current loss, and $\mathcal{L}_{\text{initial, full VQA}}$ and $\mathcal{L}_{\text{min, full VQA}}$ are the initial and minimum loss of full VQA. We use pure VQE [59] and pure QML [63] as the baseline. We use $\mathcal{L}_{\text{target}}$ at 1% relative loss for VQE and the target test accuracy (0.5% below maximum) for QML to compute the computational costs and the speedup $s$ as a metric of the performance of QuACK. Our experiments are run with Qiskit [2], Pytorch [57], Yao [47] (in Julia [7]), and Pennylane [6]. Details of the architectures and hyperparameters of our experiments are in Appendix. More ablation studies for hyperparameters are also in Appendix.

**Quantum Ising Model.** Quantum Ising model with a transverse field $h$ has the Hamiltonian $\mathcal{H} = -\sum_{i=1}^{N} Z_i \otimes Z_{i+1} - h \sum_{i=1}^{N} X_i$ where $\{I_i, X_i, Y_i, Z_i\}$ are Pauli matrices for the $i$-th qubit. On the quantum circuit, we then use the RealAmplitudes ansatz and hardware-efficient ansatz [31]

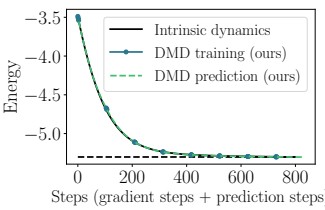
(a) Quantum natural gradients.

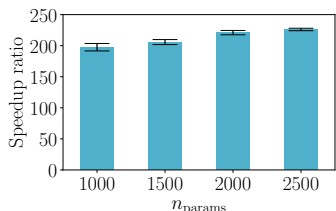
(b) Overparameterization.

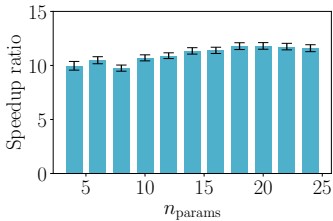
(c) Smooth optimization regimes.

Figure 2: Performance of our QuACK with the standard DMD in the following cases. (a) For quantum natural gradient, with short training (4 steps per piece) and long prediction (40 steps per piece), DMD accurately predicts the intrinsic dynamics of quantum optimization, and QuACK has 20.18x speedup. (b) In the overparameterization regime, QuACK has >200x speedup with 2-5 qubits. (c) In smooth optimization regimes, QuACK has >10x speedup with 2-12 qubits.

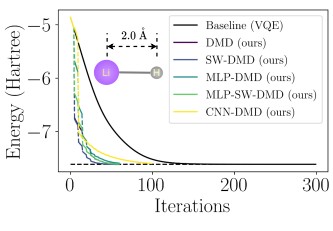
(a) LiH molecule.

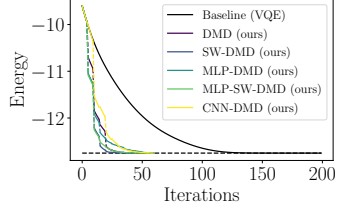
(b) Quantum Ising model.

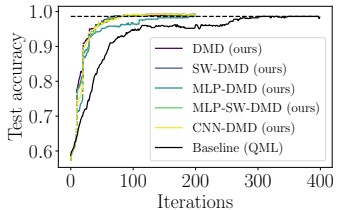
(c) Quantum machine learning.

Figure 3: Experimental results for (a) LiH molecule with 10 qubits using Adam (b) Quantum Ising model with 12 qubits using Adam (c) test accuracy of binary classification in QML. The solid piecewise curves are true gradient steps, and the dashed lines connecting them indicate when the DMD prediction is applied to find $\boldsymbol{\theta}(t_{\mathrm{opt}})$ in our controlled scheme. Our QuACK with all the DMD methods bring acceleration, with maximum speedups (a) 4.63x (b) 3.24x (c) 4.38x.

which then define $\psi_{\boldsymbol{\theta}}$, and then $\mathcal{L}(\boldsymbol{\theta}) = \langle \psi_{\boldsymbol{\theta}} | \mathcal{H} \psi_{\boldsymbol{\theta}} \rangle$. We implement the VQE algorithm and QuACK for $h = 0.5$ using gradient descent, quantum natural gradient, and Adam optimizers.

**Quantum Chemistry.** Quantum chemistry is of great interest as an application of quantum algorithms, and has a Hamiltonian $\mathcal{H} = \sum_{j=0}^{n_P} h_j P_j$ with $n_P$ polynomial in $N$, where $P_j$ are tensor products of Pauli matrices, and $h_j$ are the associated real coefficients. The loss function of quantum chemistry is typically more complicated than the quantum Ising model. We explore the performance of QuACK by applying it to VQE for the 10-qubit Hamiltonian of a LiH molecule with a interatomic distance 2.0 Å provided in Pennylane, with Adam and the RealAmplitudes ansatz.

**Quantum Machine Learning.** In addtion to VQE, we consider the task of binary classification on a filtered MNIST dataset with samples labeled by digits "1" and "9". We use an interleaved block-encoding scheme for QML, which is shown to have generalization advantage [29, 13, 40, 63] and recently realized in experiment [62]. We use a 10-qubit quantum circuit with stochastic gradient descent for QML and QuACK, and a layerwise partitioning [16] in $\boldsymbol{\theta}$ for neural DMD. Similar to the target loss, we set the target test accuracy as 0.5% below the maximum test accuracy to measure the closeness to the best test accuracy during the QML training, and use the ratios of cost between pure QML and QuACK achieving the target test accuracy as the speedup ratio.

## 6.2 Accurate Prediction for Quantum Natural Gradients by QuACK

In Figure 2a, we present the 5-qubit quantum Ising model results (learning rate 0.001) with the standard DMD method in the QuACK framework. We observe that the DMD prediction is almost perfect, i.e. $a \approx 1$. The speedup in Figure 2a is 20.18x close to 21.19x, the theoretical speedup from the upper bound in Eq. 13 under $a = 1$. This experiment shows (1) the success of DMD in predicting the dynamics (2) the power of QuACK of accelerating VQA (3) the precision of our complexity and speedup analysis in Sec. 5.2. We show 10-qubit results for all DMD methods in the Appendix.

| Noise System | $n_{\mathrm{shots}}$ | Speedup | | | | |
|---|---|---|---|---|---|---|
| | | DMD (ours) | SW-DMD (ours) | MLP-DMD (ours) | MLP-SW-DMD (ours) | CNN-DMD (ours) |
| 10-qubit shot noise | 100 | 5.51x | **5.54x** | 3.25x | 3.23x | 1.99x |
| | 1,000 | 3.20x | **4.36x** | 2.38x | **4.36x** | 2.54x |
| | 10,000 | 1.59x | **3.49x** | 1.59x | 2.41x | 1.96x |
| 5-qubit shot noise | 100 | 1.50x | 2.64x | 1.83x | **4.57x** | 1.47x |
| | 1,000 | 1.39x | **3.84x** | 1.45x | 2.25x | 1.95x |
| | 10,000 | 1.49x | **3.43x** | 1.80x | 2.00x | 1.78x |
| 5-qubit FakeLima | 100 | **2.44x** | 2.24x | 2.43x | 2.15x | 2.34x |
| | 1,000 | 1.50x | **2.61x** | 2.07x | 2.41x | 1.84x |
| | 10,000 | 1.48x | 2.32x | 1.93x | **2.56x** | 1.92x |
| 5-qubit FakeManila | 100 | 2.14x | 2.51x | 2.54x | **2.91x** | 1.25x |
| | 1,000 | 1.49x | 2.02x | 1.90x | **2.09x** | 1.89x |
| | 10,000 | 1.95x | 2.13x | 2.13x | **2.27x** | 1.82x |

Table 1: Speedup ratios of our QuACK with all DMD methods for various noise systems.

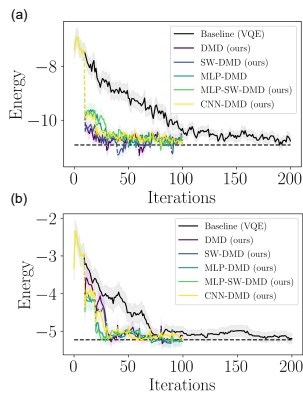

Figure 4: Noisy quantum optimization with $n_{\mathrm{shots}} = 100$. (a) 10-qubit shot noise system (b) 5-qubit FakeManila .

## 6.3 More than 200x Speedup near the Overparametrization Regime by QuACK

As it is used by the overparameterization theory in You et al. [87], we use the 250-layer hardware-efficient ansatz [31] on the quantum Ising model with gradient descent, with numbers of qubits $N = 2, 3, 4, 5$ so $n_{\mathrm{params}} = 1000, 1500, 2000, 2500$, all in the regime of $n_{\mathrm{params}} \geq (2^N)^2$, near the overparameterization regime [37]. In addition, we use a small learning rate 5e-6 so that the dynamics in the $\boldsymbol{\theta}$-space is approximately linear dynamics. We expect QuACK with the standard DMD to have good performance in this regime. In Figure 2b, for each $n_{\mathrm{params}}$, we randomly sample 10 initializations of $\boldsymbol{\theta}$ to calculate mean values and errorbars and obtain >200x speedup in all these cases. The great acceleration from our QuACK is an empirical validation and an application of the overparameterization theory for quantum optimization. On the other hand, the overparameterization regime is difficult for near-term quantum computers to realize due to the large number of parameters and optimization steps, and our QuACK makes its realization more feasible.

## 6.4 More than 10x Speedup in Smooth Optimization Regimes by QuACK

For the quantum Ising model with gradient descent and 2-layer RealAmplitudes, rather than the overparameterization regime, we consider a different regime with lower $n_{\mathrm{params}}$ with learning rate 2e-3 so that the optimization trajectory is smooth and the standard DMD is still expected to predict the dynamics accurately for a relatively long length of time. With the number of qubits $N \in [2, 12]$ ($n_{\mathrm{params}} \in [4, 24]$) and 100 random samples for initialization of $\boldsymbol{\theta}$ for each $n_{\mathrm{params}}$, in Figure 2c, our QuACK achieves >10x speedup in all these cases. This shows the potential of our methods for this regime with fewer parameters than overparameterization, which is more realistic on near-term quantum hardware.

## 6.5 More than 3x Speedup in Non-smooth Optimization by QuACK

We further demonstrate speedup from QuACK in the non-smooth optimization regimes with learning rate 0.01 for examples of quantum Ising model, quantum chemistry, and quantum machine learning with performance shown in Figure 3 with numerical speedups in Appendix. Only the gradient steps are plotted, but the computational cost of QuACK prediction steps are also counted when computing speedup. All the 5 DMD methods are applied to all the examples. Our QuACK with all the DMD methods accelerates the VQA, with maximum speedups (a) LiH molecule: 4.63x (b) quantum Ising model: 3.24x (c) QML: 4.38x. These applications are of broad interest across different fields and communities and show that our QuACK works for a range of loss functions and tasks.

### 6.6 2x to 5.5x Speedup from Ablation the Robustness of QuACK to Noise

Near-term quantum computers are imperfect and have two types of noise: (1) shot noise from the probabilistic nature of quantum mechanics, which decreases as $1/\sqrt{n_{\text{shots}}}$, (2) quantum noise due to the qubit and gate errors which cannot be removed by increasing $n_{\text{shots}}$. Therefore, we consider two categories of ablation studies (1) with shot noise only (2) with both shot noise and quantum noise. We apply our QuACK with all 5 DMD methods to 4 types of noise systems: 10-qubit and 5-qubit systems with only shot noise, FakeLima, and FakeManila. The latter two are noise models provided by Qiskit to mimic 5-qubit IBM real quantum computers, Lima and Manila, which contain not only shots noise but also the machine-specific quantum noise. We use the quantum Ising model with Adam and show generic dynamics in Figure 4 with statistical error from shots as error bands. The fluctuation and error from baseline VQE in Figure 4(a) 100-shot 10-qubit system are less than Figure 4(b) 100-shot FakeManila. Our QuACK works well in (a) up to 5.54x speedup and in (b) up to 2.44x speedup. In all examples, we obtain speedup and show them in Table 1, which demonstrate the robustness of our QuACK in realistic setups with generic noisy conditions. We have also implemented an experiment to accelerate VQE on the real IBM quantum computer Lima with results in Appendix.

## 7   Conclusion

We developed QuACK, a novel algorithm that accelerates quantum optimization. We derived QuACK from connecting quantum natural gradient theory, overparameterization theory, and Koopman operator learning theory. We rigorously tested QuACK's performance, robustness, spectral stability, and complexity on both simulated and real quantum computers across a variety of scenarios. Notably, we observed orders of magnitude speedups with QuACK, achieving over 200 times faster in overparameterized regimes, 10 times in smooth regimes, and 3 times in non-smooth regimes. This research highlights the significant potential of Koopman operator theory for accelerating quantum optimization and lays the foundation for stronger connections between machine learning and quantum optimization. We elaborate more on the broader impact of our work in the Appendix.

### Acknowledgement

The authors acknowledge helpful discussions with Hao He, Charles Roques-Carmes, Eleanor Crane, Nathan Wiebe, Zhuo Chen, Ryan Levy, Lucas Slattery, Bryan Clark, Weikang Li, Xiuzhe Luo, Patrick Draper, Aida El-Khadra, Andrew Lytle, Yu Ding, Ruslan Shaydulin, Yue Sun. DL, RD and MS acknowledge support from the NSF AI Institute for Artificial Intelligence and Fundamental Interactions (IAIFI). DL is supported in part by the Co-Design Center for Quantum Advantage (C2QA). JS acknowledges support from the U.S. Department of Energy, Office of Science, Office of High Energy Physics QuantISED program under an award for the Fermilab Theory Consortium "Intersections of QIS and Theoretical Particle Physics". This material is also in part based upon work supported by the Air Force Office of Scientific Research under the award number FA9550-21-1-0317. We acknowledge the use of IBM Quantum services for this work. The views expressed are those of the authors, and do not reflect the official policy or position of IBM or the IBM Quantum team. In this paper we used *ibmq_lima*, which is one of the IBM Quantum Falcon Processors.

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

# Appendix

The structure of our Appendix is as follows. Appendix A gives an introduction to quantum mechanics and the overparameterization theory as preliminaries of our paper. Appendix B provides more details of our QuACK framework introduced in Sec. 4 of main text. Appendix D provides proofs of our theoretical results in Sec. 5 of main text. Appendix E discusses detailed information on Sec. 6 Experiments of main text. Appendix F contains additional information.

## A    Preliminaries

### A.1    Introduction to Quantum Mechanics for Quantum Computation

We introduce quantum mechanics for quantum computation in this section. In quantum mechanics, the basic element to describe the status of a system is a quantum state (or a wave function) $\psi$. A pure quantum state is a vector in a Hilbert space. We can also use the Dirac notation for quantum states $\langle\psi|$ and $|\psi\rangle$ to denote the row and column vectors respectively. By convention, $\langle\psi|$ is the Hermitian conjugate (composition of complex conjugate and transpose) of $|\psi\rangle$.

In a quantum system with a single qubit, there are two orthonormal states $|0\rangle$ and $|1\rangle$, and a generic quantum state is spanned under this basis as $|\psi\rangle = c_0|0\rangle + c_1|1\rangle$ where $c_0, c_1 \in \mathbb{C}$. Therefore, $\psi$ itself is in a 2-dimensional Hilbert space as $\psi \in \mathbb{C}^2$. For a quantum mechanical system with $N$ qubits, the basis of quantum state contains tensor products (Kronecker products) of the single-qubit bases, $i.e.$, $|b_1\rangle \otimes |b_2\rangle \otimes \cdots \otimes |b_N\rangle$, where $b_i \in \{0, 1\}$ ($i = 0, 1, \cdots, N$). There are in total $2^N$ basis vectors, and $\psi$ is a linear combination of them with complex coefficents. Therefore, $\psi$ for an $N$-qubit system is in a $2^N$-dimensional complex-valued Hilbert space $\mathbb{C}^{2^N}$. For a normalized quantum state, we further require the constraint $\|\psi\|_2^2 = 1$ where $\|\cdot\|_2$ is the $l_2$-norm.

For the $N$-qubit system, the Hamiltonian $\mathcal{H}$ is a Hermitian $2^N \times 2^N$-matrix acting on the quantum state (wave function) $\psi$. The energy of $\psi$ is given by $\mathcal{L}(\psi) = \langle\psi|\mathcal{H}\psi\rangle$, which is the inner product between $\langle\psi|$ (a row vector, the Hermitian conjugate of $|\psi\rangle$) and $|\mathcal{H}\psi\rangle$ (matrix multiplication between the matrix $\mathcal{H}$ and the column vector $|\psi\rangle$).

Pauli matrices (including the identity matrix $I$ by our convention) are the $2 \times 2$ matrices

$$I = \begin{bmatrix} 1 & 0 \\ 0 & 1 \end{bmatrix}, \quad X = \begin{bmatrix} 0 & 1 \\ 1 & 0 \end{bmatrix}, \quad Y = \begin{bmatrix} 0 & -i \\ i & 0 \end{bmatrix}, \quad Z = \begin{bmatrix} 1 & 0 \\ 0 & -1 \end{bmatrix}. \tag{4}$$

The Pauli matrices can act on a single-qubit (2-dimensional) state through matrix multiplication. For the $N$-qubit system, we need to specify which qubit the Pauli matrix is acting on by the subscript $i$ for the $i$-th qubit. For example, $X_2$ denotes the $X$ Pauli matrix acting on the second qubit. To have a complete form of a $2^N \times 2^N$-matrix acting on the $N$-qubit system, we need tensor products of Pauli matrices, $i.e.$, Pauli strings. For example, $I_1 \otimes Z_2 \otimes Z_3 \otimes I_4$ is a $2^4 \times 2^4$-matrix acting on a 4-qubit system. Conventionally, without ambiguity, when writing a Pauli string matrix, we can omit the identity matrices, and it will be equivalent to write the above Pauli string matrix as $Z_2 \otimes Z_3$ (which, as an example, appears in the Hamiltonian of the quantum Ising model: $\mathcal{H} = -\sum_{i=1}^{N} Z_i \otimes Z_{i+1} - h\sum_{i=1}^{N} X_i$). Pauli string can serve as the basis of writing the Hamiltonian, which is manifested by format of the Hamiltonian used in quantum chemistry: $\mathcal{H} = \sum_{j=0}^{n_P} h_j P_j$ with $n_P$ polynomial in $N$, where $P_j$ are pauli strings, and $h_j$ are the associated real coefficients.

In quantum computation, we have quantum gates parametrized by $\theta$. A common example is the single-qubit rotational gates $R_X(\theta)$, $R_Y(\theta)$, $R_Z(\theta)$, $2 \times 2$-matrices defined as

$$R_X(\theta) = \exp\left(-i\theta X/2\right) = \begin{bmatrix} \cos\left(\theta/2\right) & -i\sin\left(\theta/2\right) \\ -i\sin\left(\theta/2\right) & \cos\left(\theta/2\right) \end{bmatrix}, \tag{5}$$

$$R_Y(\theta) = \exp\left(-i\theta Y/2\right) = \begin{bmatrix} \cos\left(\theta/2\right) & -\sin\left(\theta/2\right) \\ \sin\left(\theta/2\right) & \cos\left(\theta/2\right) \end{bmatrix}, \tag{6}$$

$$R_Z(\theta) = \exp\left(-i\theta Z/2\right) = \begin{bmatrix} \exp\left(-i\theta/2\right) & 0 \\ 0 & \exp\left(i\theta/2\right) \end{bmatrix}, \tag{7}$$

where $\theta \in \mathbb{R}$ is the rotational angle as the parameter of a rotational gate. In a quantum circuit, there can be a number of rotational gates acting on different qubits, and the parameters $\theta$ in them can be chosen independently. All these components of $\theta$ then get combined into a vector $\boldsymbol{\theta} \in \mathbb{R}^{n_{\mathrm{params}}}$. Note that the space of $\boldsymbol{\theta}$ is different from the space of $\psi$.

The different qubits in the quantum circuit would still be disconnected with only single-qubit rotational gates. To connect the different qubits, the 2-qubit controlled gates, including controlled-$X$, controlled-$Y$, and controlled-$Z$, (with no parameter) are needed

$$CX = \begin{bmatrix} 1 & 0 & 0 & 0 \\ 0 & 0 & 0 & 1 \\ 0 & 0 & 1 & 0 \\ 0 & 1 & 0 & 0 \end{bmatrix}, \tag{8}$$

$$CY = \begin{bmatrix} 1 & 0 & 0 & 0 \\ 0 & 0 & 0 & -i \\ 0 & 0 & 1 & 0 \\ 0 & i & 0 & 0 \end{bmatrix}, \tag{9}$$

$$CZ = \begin{bmatrix} 1 & 0 & 0 & 0 \\ 0 & 1 & 0 & 0 \\ 0 & 0 & 1 & 0 \\ 0 & 0 & 0 & -1 \end{bmatrix}. \tag{10}$$

They are $2^2 \times 2^2$-matrices that act on 2 qubits $i$, $j$ out of the $N$ qubits.

**RealAmplitudes ansatz.** The RealAmplitudes ansatz is a quantum circuit that can prepare a quantum state with real amplitudes. The implementation of RealAmplitudes in Qiskit is a parameterized $R_Y$-layer acting on all the $N$ qubits and then repetitions of an entanglement layer of $CX$ and a parameterized $R_Y$-layer. The number of reptitions is denoted as "reps". The total number of entanglement layers (with no parameters) is reps, and the total number of parameterized $R_Y$-layers is reps $+ 1$. The total number of parameters is $n_{\mathrm{params}} = N(\mathrm{reps} + 1)$. The entanglement layers in our experiment are chosen as a circular configuration.

**Hardware-efficient ansatz.** We use the hardware-efficient ansatz following the convention of You et al. [87]. In each building block, there are an $R_X$-layer, an $R_Y$-layer, and the $CZ$-entanglement layer at odd links and then even links (acting on the state in this described order). Then there are in total repetitions of "depth" of such building blocks. In the end, there are $2N \cdot \mathrm{depth}$ parameters.

## A.2 Overparameterization Theory

Recent development on overparameterization theories has shown that VQA with gradient-based optimization has convergence guarantee [37, 44, 87]. Since there are different assumptions behind different overparameterization theories, here we provide a brief introduction to the overparameterization theory in You et al. [87], and we refer the readers to the original paper for more details. The important insight of the overparameterization theory comes from that in large-$n_{\mathrm{params}}$ regime with proper learning rate, the dynamics of the wave function $|\psi_{\boldsymbol{\theta}}(t)\rangle$ from the parameterized quantum circuit is described by the following equation

$$\frac{d|\psi_{\boldsymbol{\theta}}(t)\rangle}{dt} = -[\mathcal{H}, |\psi_{\boldsymbol{\theta}}(t)\rangle \langle\psi_{\boldsymbol{\theta}}(t)|]\,|\psi_{\boldsymbol{\theta}}(t)\rangle + \delta f(|\psi_{\boldsymbol{\theta}}(t)\rangle), \tag{11}$$

where $\mathcal{H}$ is the Hamiltonian in a VQE task, and $[\ ,\ ]$ is the commutator. The first term in the R.H.S. is the Riemannian gradient descent over the normalized wave function manifold with energy as the loss function and converges linearly to the ground state. This is also similar to the case of the quantum natural gradient. The second term in the R.H.S. is a small perturbation function $\delta f$ of $|\psi_{\boldsymbol{\theta}}(t)\rangle$ (see the detailed form in You et al. [87]). It has been shown that in the overparameterization regime, the equation can still converge to the ground state under such perturbation. Similar to the argument in the quantum natural gradient case, the overparameterization theory provides an effective structure for Koopman learning with approximate linear dynamics.

# B   Details of Sec. 4 QuACK - Quantum-circuit Alternating Controlled Koopman Operator Learning

## B.1   Futher details of Neural DMD

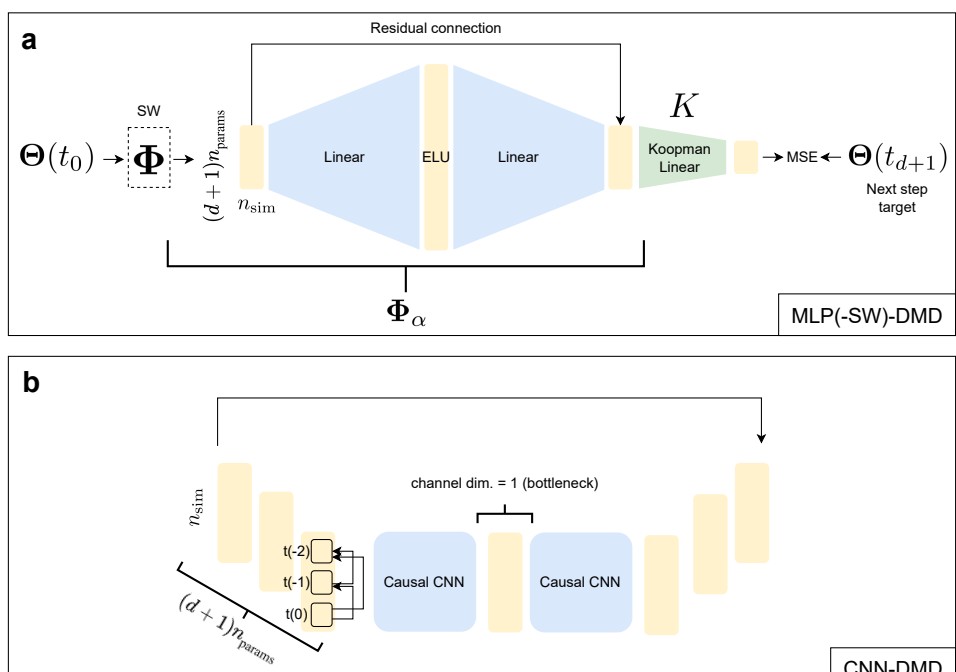

Figure 5: Neural network architectures for our neural DMD approaches. The factor $(d+1)$ in the dimension $(d+1)n_{\text{params}}$ is due to the sliding window embedding $\mathbf{\Phi}$. (a) MLP bottleneck architecture with MSE loss for training. (b) CNN bottleneck architecture that operates on simulations as temporal and parameters as channel dimensions.

The neural network architectures for our neural DMD, *i.e.*, MLP-DMD, MLP-SW-DMD, and CNN-DMD, are shown in Figure 5. For the CNN-DMD, we use an expansion ratio of 1 in the experiments.

**Optimization.** The parameters $K$ and $\alpha$ are trained jointly on $\arg\min_{K,\alpha}\|\mathbf{\Theta}(t_{d+1}) - K\mathbf{\Phi}_\alpha(\mathbf{\Theta}(t_0))\|_F$, from scratch with every new batch of optimization history by using the Adam optimizer for 30k steps with 9k steps of linear warmup from 0 to 0.001 and then cosine decay back to 0 at step 30k. We use the MSE loss, which minimizes the Frobenius norm.

**Activation**   Along with ELU, we also explored cosine, ReLU and tanh activations. We found ELU to be the best, likely because: tanh suffers from vanishing gradients, ReLU biases to positive numbers (while input phases quantum circuit parameters $\boldsymbol{\theta}$ are unconstrained) and cosine is periodic.

**The importance of learning rate scheduler.**   We train the neural network $\mathbf{\Phi}_\alpha$ from scratch every time we obtain optimization history as training data. It is important that we have a stable learning with well converging neural network at every single Koopman operator fitting stage. For that purpose, we found it vital to use a cosine-decay scheduler with a linear warmup, which is typically useful in the computer vision literature [46]. Namely, for the first 9k steps of the neural network optimization, we linearly scale the learning rate from 0 to 0.001, and then use a cosine-decay from 0.001 to 0 until the final step at 30k. In Sec. 6 for quantum machine learning, we use 160k training steps for CNN to achieve a better performance.

**The importance of residual connections.**   In our work we use a residual connection in order to make DMD as a special case of the neural DMD parameterization. The residual connection is indeed

very useful, as it is driven by the Koopman operator learning formulation. Namely, the residual connection makes it possible for the encoder to learn the identity. If the encoder becomes the identity, then MLP(-SW)-DMD or CNN-DMD become vanilla (SW-)DMD.

**The procedure for making predictions using neural DMD.** Sec. 4 provides the general objective $\arg\min_{K,\alpha} \|\Theta(t_{d+1}) - K\Phi_\alpha(\Theta(t_0))\|_F$ for training the neural-network DMD including MLP-DMD, CNN-DMD, and MLP-SW-DMD. First, for MLP-DMD (and CNN-DMD with no sliding window), we take $d = 0$ with no sliding window, so the objective gets reduced to $\arg\min_{K,\alpha} \|\Theta(t_1) - K\Phi_\alpha(\Theta(t_0))\|_F$. In the phase of training, the optimization history $\theta(t_0), \theta(t_1), ..., \theta(t_{m+1})$ is first concatenated into $\Theta(t_0)$ and $\Theta(t_1)$ in the same way as in SW-DMD using $\Theta(t_k) = [\theta(t_k)\ \theta(t_{k+1}) \cdots \theta(t_{k+m})]$. In every column, $\Theta(t_1)$ is one iteration ahead of $\Theta(t_0)$ in the future direction. Then, in training, the operator $\Phi_\alpha = \mathrm{NN}_\alpha$, as a neural network architecture, acts only on $\Theta(t_0)$ not on $\Theta(t_1)$. $K$ is a square Koopman matrix that denotes a forward dynamical evolution from $\Phi_\alpha\Theta(t_0)$ directly to $\Theta(t_1)$, rather than from $\Phi_\alpha\Theta(t_0)$ to $\Phi_\alpha\Theta(t_1)$. Likewise, in the phase of inference for predicting the future of $\theta$ beyond $t_{m+1}$, we apply the operator $K\Phi_\alpha$ repeatedly using the evolution $\Theta(t_{k+1}) = (K\Phi_\alpha)^k\Theta(t_1)$, without an explicit inversion of the operator $\Phi_\alpha$ to bring back the original representation. Next, for MLP-SW-DMD, we need to put $d$ back to the equations and make the operator $\Phi_\alpha = \mathrm{NN}_\alpha \circ \Phi$ a composition of the neural network architecture $\mathrm{NN}_\alpha$ and the sliding window embedding $\Phi$. The Koopman operator $K$ of MLP-SW-DMD has the same dimension as $K$ of SW-DMD, which is non-square. The procedure of updating $\Phi(\Theta(t))$ by adding the latest data and removing the oldest data is the same as in SW-DMD described in Sec. 4. The only difference between MLP-SW-DMD and SW-DMD is the additional neural network architecture $\mathrm{NN}_\alpha$ in MLP-SW-DMD. The only difference between MLP-SW-DMD and MLP-DMD is the additional sliding window embedding $\Phi$ in MLP-SW-DMD. The same relationship holds true for CNN-DMD between the cases of $d = 0$ and $d > 0$.

## B.2 The Alternating Controlled Scheme

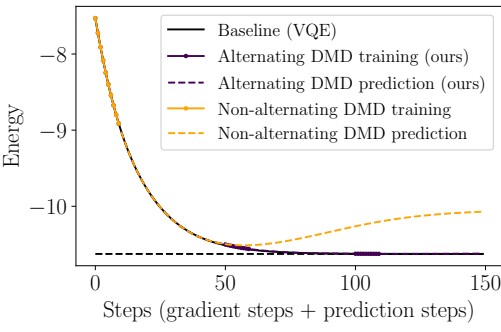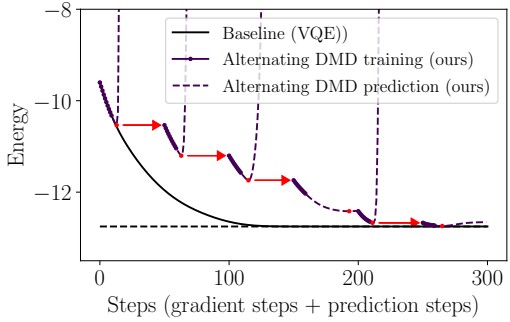

(a) Quantum natural gradient results using alternating and non-alternating traditional DMD.

(b) Adam results using traditional DMD with the alternating scheme.

Figure 6: Motivating examples for the alternating scheme and nonlinear embeddings. The results are from the 10-qubit quantum Ising model. Solid and dashed parts indicate true gradient steps and prediction steps respectively. In the starting regime of (a) where the lines almost overlap, the first 10 steps are solid for the DMD lines. In (b), the red points mark the optimal parameters for each piece of DMD predictions. The next piece of VQE starts from the optimal parameters rather than the last parameters, as is indicated by the red arrows, to guarantee the decrease of energy.

The Koopman operator theory is a powerful framework for understanding and predicting nonlinear dynamics through linear dynamics embedded into a higher dimensional space. By viewing parameter optimization on quantum computers as a nonlinear dynamical evolution in the parameter space, we connect gradient dynamics in quantum optimization to the Koopman operator theory. In particular, the quantum natural gradient helps to provide a effective structure of parameter embedding into a higher-dimensional space, related to imaginary-time evolution. However, for quantum optimization, using the standard dynamic mode decomposition (DMD) that assumes linear dynamics in optimization directly has the following issues demonstrated in Figure 6 of the 10-qubit quantum Ising model with $h = 0.5$ learning rate 0.01. (1) In Figure 6a, the case of natural gradient, although the dynamics is

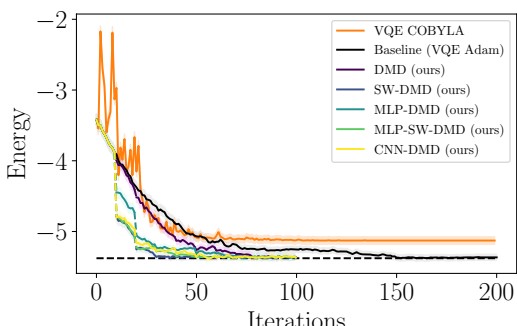

Figure 7: 5-qubit shot noise, $n_{\text{shots}} = 1,000$. VQE with COBYLA is plotted in orange in addition to the baseline VQE with Adam and all the DMD methods in our QuACK. The COBYLA gets trapped.

approximately linear (as we discuss in main text), without using an alternating scheme, the DMD prediction is only good up to about 50 steps and starts to differ from the baseline VQE. This issue is much mitigated by adopting our alternating scheme by running follow-up true gradient steps. (2) In Figure 6b, even with the alternating scheme, in the case of Adam where the dynamics is less linear, the standard DMD method can start to diverge quickly. To mitigate this issue, we control our alternating scheme by starting the follow-up true gradient steps from the optimal parameters in prediction. In addition, we propose to use the sliding-window embedding and neural-networks to better capture the nonlinearity of dynamics.

From Theorem 5.1, the asymptotic behavior of DMD prediction for quantum optimization is trivial or unstable. This is manifested in Figure 6b, as the energy of DMD prediction diverges as the prediction time increases.

## C    An Example Comparing Gradient-Based and Gradient-Free Methods

While gradient-based methods and gradient-free methods are both applicable types of optimizers for VQAs, gradient-based methods use gradient information and thus can know better about the local geometry [39], especially with quantum Fisher information through quantum natural gradient. Particularly, the gradient-based method has been shown to converge for quantum optimization [74, 87].

In Figure 7, we show an example of the quantum Ising model with shot noise. While the gradient-based VQE with Adam finds a lower energy than COBYLA with `rhobeg=1.0`, as COBYLA gets trapped. The results of Adam and QuACK are from Figure 12e. Our QuACK further improve upon Adam.

## D    Proofs of Theoretical Results in Sec. 5

**Corollary 5.3.** *QuACK achieves superior performance over the asymptotic DMD prediction.*

*Proof.* As it is shown in Theorem 5.1, asymptotic DMD prediction could lead to trivial or unstable solution. Since QuACK employs the alternating controlled scheme, Theorem. 5.2 shows that QuACK obtains the optimal prediction in each iteration. It implies that even the asymptotic prediction converges to trivial solution or oscillates, QuACK can take the best prediction among the sufficiently long DMD predictions to achieve superior performance. □

**Corollary D.1.** *QuACK is guaranteed to converge for convex optimization on function $f : \mathbb{R}^m \to \mathbb{R}$ that is convex and differentiable with gradient that is Lipschitz continuous of positive Lipschitz constant.*

*Proof.* Since $f$ has gradient that is Lipschitz continuous with positive Lipschitz constant, there exists some $L > 0$ such that $||f'(x) - f'(y)||_2 \le L||x - y||_2$. It has been shown that by choosing a fixed

step size $\eta = 1/L$, the gradient descent is guaranteed to converge [56]. Each gradient step update from $x_k$ is guaranteed to have $f(x_{k+1}) < f(x_k) - \frac{1}{2L}||f'(x_k)||^2$ and after $n$ gradient steps from $x^{(0)}$

$$f(x^{(n)}) - f(x^*) \leq \frac{||x^{(0)} - x^*||_2^2}{2n\eta} \tag{12}$$

where $x^*$ is the optimal solution. Since Theorem 5.2 shows that QuACK yields an equivalent or lower loss than $n$-step gradient descent and we discuss above that the convex function $f$ with a positive Lipschitz constant can converge from any $x^{(0)}$ with a proper step size $\eta$, it follows that QuACK is guaranteed to converge in the same setup. □

**Theorem 5.4.** *With a baseline gradient-method, the speedup ratio $s$ is in the following range*

$$a \leq s \leq a \frac{f(p)(n_{\text{sim}} + n_{\text{DMD}})}{f(p)n_{\text{sim}} + n_{\text{DMD}}}. \tag{13}$$

*where $a := T_b/T_{Q,t}$. In the limit of $n_{\text{iter}} \to \infty$ the upper bound can be achieved.*

To achieve $\mathcal{L}_{\text{target}}$, the baseline VQA takes $T_b$ gradient steps, and QuACK takes $T_{Q,t} = T_{Q,1} + T_{Q,2}$ steps, where $T_{Q,1}$ ($T_{Q,2}$) are the total numbers of QuACK training (prediction) steps. $s$ is the speedup ratio from QuACK. $p := n_{\text{params}}$ is the number of parameters.

*Proof.* The total computational costs of the baseline and QuACK both contain two parts: classical computational cost and quantum computational cost. For the baseline gradient-based optimizer, the total computational cost is $q\, n_{\text{shots}}|B_q|f(p)T_b + c_b$, where $n_{\text{shots}}$ is the number of quantum measurements per VQA example per gradient step per parameter, $|B_q|$ is the batch size of VQA examples (*e.g.*, $|B_q|$ is the number of training examples in the batch of QML, and is the number of terms in the Hamiltonian in VQE), $q$ is the quantum computational cost per VQA example per shot per gradient step per parameter, and $c_b$ is the total classical computational cost for the baseline, from processing the input and output of the quantum computer using a classical computer. $c_b$ depends on the details of the VQA task, and for the tasks feasible on near-term quantum computers, $c_b \ll q$ and thus can be neglected since the quantum resources are much more expensive than classical resources. $f(p)$ is the optimizer factor for per gradient step evaluation on a quantum computer. For example, we have $f(p) = 2p + 1$ for parameter-shift rules and $f(p) \sim p^2 + p$ for quantum natural gradients. For QuACK, the total computational cost is $q\, n_{\text{shots}}|B_q|[f(p)T_{Q,1} + T_{Q,2}] + c_Q$, where $c_Q$ the total classical computational cost for QuACK, depending on the type of DMD method and the number of classical training steps for neural DMD, plus classical processing of the input and output of the quantum computer. Again, for near-term quantum computers, $c_Q \ll q$ for the same reason as $c_b$ and thus can be neglected. The QuACK training term has the factor $f(p)$, while the QuACK prediction term does not, because it does not involve taking gradients. The speedup ratio by QuACK is then

$$\begin{aligned}
s &= \frac{q\, n_{\text{shots}}|B_q|f(p)T_b + c_b}{q\, n_{\text{shots}}|B_q|[f(p)T_{Q,1} + T_{Q,2}] + c_Q} \\
&\approx \frac{q\, n_{\text{shots}}|B_q|f(p)T_b}{q\, n_{\text{shots}}|B_q|[f(p)T_{Q,1} + T_{Q,2}]} \\
&= \frac{f(p)T_b}{f(p)T_{Q,1} + T_{Q,2}} \\
&= a \frac{f(p)(T_{Q,1} + T_{Q,2})}{f(p)T_{Q,1} + T_{Q,2}}
\end{aligned} \tag{14}$$

When deriving Eq. 14, the common factor $q\, n_{\text{shots}}|B_q|$ in the numerator and denominator is exactly the same, so its cancellation is exact. Since QuACK training always occurs before QuACK prediction, whether $\mathcal{L}_{\text{target}}$ is achieved during QuACK training or prediction, the general case is $T_{Q,1}/T_{Q,2} \geq n_{\text{sim}}/n_{\text{DMD}}$. Therefore,

$$a \leq s \leq a \frac{f(p)(n_{\text{sim}} + n_{\text{DMD}})}{f(p)n_{\text{sim}} + n_{\text{DMD}}}. \tag{15}$$

In the limit $n_{\text{iter}} \to \infty$, the location (measured in terms of the number of QuACK steps) of the last QuACK step does not affect $T_{Q,1}/T_{Q,2}$ much, and we have $T_{Q,1}/T_{Q,2} \to n_{\text{sim}}/n_{\text{DMD}}$, so the upper bound in Eq. 13 can be achieved. □

# E Detailed Information on Sec. 6 Experiments

## E.1 Experimental Setup Details

We use the notation $T_{b,t}$ as the total number of pure VQA (baseline) gradient steps in the plots, and $n_{\mathrm{SW}}$ as the sliding-window size (equal to $d + 1$). $n_{\mathrm{iter}}$ is the number of repetitions of alternating VQA+DMD runs, which is chosen large enough to ensure convergence of the baseline VQA. When convergence is achieved, the performance of QuACK does not have a strong dependence on choice of $n_{\mathrm{iter}}$.

### E.1.1 Quantum Ising Model Setup

For the quantum Ising model Hamiltonian $\mathcal{H} = -\sum_{i=1}^{N} Z_i \otimes Z_{i+1} - h \sum_{i=1}^{N} X_i$, we use the periodic boundary condition, such that $i = 1$ and $i = N + 1$ are identified.

In Sec. 6.2 Quantum Natural Gradient, we use the circular-entanglement RealAmplitudes ansatz and reps=1 (2 layers, $2N$ parameters) with $T_{b,t} = 800$, quantum natural gradient optimizer, learning rate 0.001. For the QuACK hyperparameters, we choose $n_{\mathrm{sim}} = 4$ and $n_{\mathrm{DMD}} = 100$ with $n_{\mathrm{iter}} = 8$. The random sampling of initialization of $\boldsymbol{\theta}$ is from a uniform distribution in $[0, 1)^{n_{\mathrm{params}}}$.

In Sec. 6.3, we use the hardware-efficient ansatz with 250 layers ($500N$ parameters) with $T_{b,t} = 5000N$, gradient descent optimizer, learning rate 5e-6. For the QuACK hyperparameters, we choose $n_{\mathrm{sim}} = 4$ and $n_{\mathrm{DMD}} = 1000$ with $n_{\mathrm{iter}} = 5N$ to ensure convergence.

In Sec. 6.4, we use the circular-entanglement RealAmplitudes ansatz and reps=1 (2 layers, $2N$ parameters) with $T_{b,t} = 1500$, Adam optimizer, learning rate 2e-3. For the QuACK hyperparameters, we choose $n_{\mathrm{sim}} = 3$ and $n_{\mathrm{DMD}} = 60$ with $n_{\mathrm{iter}} = \lceil 4T_{b,t}/(n_{\mathrm{sim}} + n_{\mathrm{DMD}}) \rceil = 96$ to ensure convergence. Some examples have an earlier stop, which does not affect calculation of the speedup ratio since they has already achieve $\mathcal{L}_{\mathrm{target}}$ before 96 pieces of VQE+DMD.

In Sec. 6.5, we use the circular-entanglement RealAmplitudes ansatz for the 12-qubit quantum Ising model and reps=1 (2 layers, $2N = 24$ parameters) with $T_{b,t} = 300$ learning rate 0.01. For the QuACK hyperparameters, we choose $n_{\mathrm{sim}} = 5$ and $n_{\mathrm{DMD}} = 40$ with $n_{\mathrm{iter}} = 12$ to ensure convergence. We have the sliding-window size $n_{\mathrm{SW}} = 3$ for SW-DMD, MLP-SW-DMD, and CNN-DMD.

### E.1.2 Quantum Chemistry Setup

A molecule of LiH, depicted in Figure 3a, consists of a lithium atom and a hydrogen atom separated by a distance. We use the quantum chemistry module from Pennylane [6] to obtain the 10-qubit Hamiltonian of LiH at an interatmoic separation 2.0 with 5 active orbitals. We use the circular-entanglement RealAmplitudes ansatz and reps=1 (2 layers, 20 parameters). The optimizer is Adam with the learning rate 0.01. We perform $T_{b,t} = 1000$ baseline VQE iterations. We choose $n_{\mathrm{sim}} = 5$, $n_{\mathrm{DMD}} = 40$, $n_{\mathrm{iter}} = 12$. The sliding-window size is $n_{\mathrm{SW}} = 3$ for SW-DMD, MLP-SW-DMD, and CNN-DMD.

### E.1.3 QML Architecture and Training Details

In our QML example in Sec. 6, the quantum computer has $N = 10$ qubits. The architecture is shown in Figure 8. For the MNIST binary classification task, each image is first downsampled to $16 \times 16$ pixels, and then used as an input $\boldsymbol{x} \in [0, 1]^{256}$ that is fed into an interleaved encoding quantum gate,. The parameters $\boldsymbol{\theta}$ are also encoded in the interleaved encoding quantum gate. Then the quantum measurements are used as the output for computing the cross-entropy loss.

The interleaved encoding gate consists of 9 layers. Each layer has a rotational layer and a linear entanglement layer. On the rotational layer, each qubit has three rotational gates $R_X, R_Z, R_X$ in a sequence. Therefore, each layer has 30 rotational angles, and the whole QML architecture has 270 rotational angles, *i.e.*, $n_{\mathrm{params}} = 270$. Since each input example $\boldsymbol{x}$ is 256-dimensional, we only use the first 256 rotational angles to encode the input data. The parameters $\boldsymbol{\theta}$ are also encoded in the rotational angles such that the angles are $\boldsymbol{x} + \boldsymbol{\theta}$.

Quantum measurements, as the last part of the quantum circuit, map the quantum output to the classical probability data. In each quantum measurement, each qubit is in the 0-state or the 1-state, and the probability for the qubit to be in the 0-state is between 0 and 1. We regard this probability as

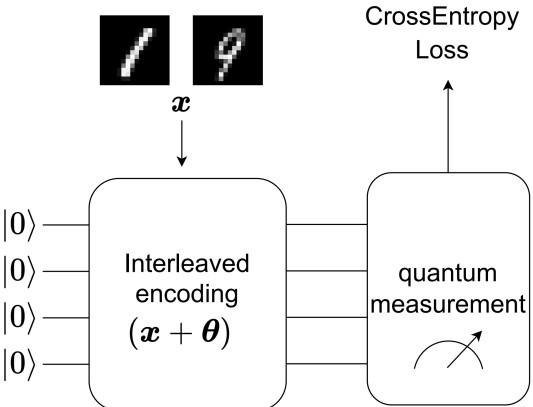

Figure 8: Quantum machine learning architecture with interleaved encoding of 10-qubit quantum circuit for a binary classification task on MNIST.

the probablity for the image to be a digit "1". We only use the probability on the 5th qubit ($i = 5$) as the output and compute the cross-entropy loss with the labels.

In the phase of training, all the training examples share the same $\boldsymbol{\theta}$ but have different $\boldsymbol{x}$, and we optimize the final average loss with respect to $\boldsymbol{\theta}$ as

$$\boldsymbol{\theta}^* = \arg\min_{\boldsymbol{\theta}} \frac{1}{n_{\text{train}}} \sum_{i=1}^{n_{\text{train}}} \mathcal{L}(\boldsymbol{x}_i^{\text{train}}; \boldsymbol{\theta}), \tag{16}$$

in the case of $n_{\text{train}}$ training examples. The combination $\boldsymbol{x}_i^{\text{train}} + \boldsymbol{\theta}$ is entered into the quantum circuit rotational angles, and we need to build $n_{\text{train}}$ separate quantum circuits (which can be built either sequentially or in parallel). The interleaved encoding gate as a whole can be viewed as a deep layer, and serve dual roles of encoding the classical data $\boldsymbol{x}$ and containing the machine learning parameters $\boldsymbol{\theta}$. In the phase of inference, with the optimal parameter $\boldsymbol{\theta}^*$, we feed each test example $\boldsymbol{x}_i^{\text{test}}$ into the quantum circuit as a combination $\boldsymbol{x}_i^{\text{test}} + \boldsymbol{\theta}^*$ and measure the output probability to compute the test accuracy.

We use 500 training examples and 500 test examples. During training, we use the stochastic gradient descent optimizer with the batch size 50 and learning rate 0.05. The full QML training has $T_{b,t} = 400$ iterations. We choose $n_{\text{sim}} = 10$, $n_{\text{DMD}} = 20$, and $n_{\text{SW}} = 6$ for SW-DMD and MLP-SW-DMD.

In neural DMD including MLP-DMD, MLP-SW-DMD, CNN-DMD, we use layerwise partitioning that groups $\boldsymbol{\theta}$ by layers in the encoding gate. There are 9 groups with group size 30. We perform neural DMD for each group, so that the number of parameters in the neural networks for DMD is not too large. After predicting each group separately, we combine all groups of $\boldsymbol{\theta}$ to evaluate the loss and accuracy. In CNN-DMD for QML, we use 160k steps for sufficient CNN training. For the stability of CNN-DMD in presence of the layerwise partitioning, we choose $n_{\text{SW}} = 1$ for CNN-DMD.

### E.2   Wallclock Time in Classical Simulations

We define the wallclock time as the time of DMD learning plus the time of quantum optimization. The DMD learning on classical computers is indeed very efficient (for each piece, only milliseconds for DMD and SW-DMD, and less than 1 minute for neural DMD), and thus it is not the bottleneck. The main bottleneck is the time of quantum optimization, and our QuACK is designed to speed up this. In our work, the quantum optimization is simulated by classical computers. The simulations on a classical computer (a single CPU) take the time at orders from minutes to an hour depending on the details of the task and hyperparameters. As a representative example, for the 12-qubit Ising model with Adam in Figure 3b of Sec. 6.5, the time costs of simulations (with the parameter-shift gradient updates explicitly to mimic the actual quantum optimization) are listed in Table 2. Thanks to the acceleration by our QuACK, we can see the benefit of having less wallclock time by our various DMD methods compared to the baseline VQE. We note that these wallclock times are for classical simulations of quantum optimization. For actual quantum optimization, it is expected that

| Method | Wallclock time (seconds) |
|---|---|
| VQE (baseline) | 2866 |
| DMD (ours) | 646 |
| SW-DMD (ours) | 663 |
| MLP-DMD (ours) | 818 |
| SW-MLP-DMD (ours) | 849 |
| CNN-DMD (ours) | 828 |

Table 2: Wallclock time costs of simulations with the parameter-shift rule explicitly implemented on a classical computer with a single CPU. Our various QuACK methods are faster than the baseline because they use fewer gradient steps.

our approach can gain much more benefit, since the dominant cost is from each gradient step of quantum optimization. The wallclock time saved on an actual quantum computer will be proportional to the speedup factor multiplying the actual time of each iteration in quantum optimization. This will help us to save more resources since the actual quantum optimization time is more expensive than the classical simulations of the quantum optimization.

### E.3  Speedup in Sec. 6.5

| | Speedup | | | | |
|---|---|---|---|---|---|
| Task | DMD | SW-DMD | MLP-DMD | MLP-SW-DMD | CNN-DMD |
|---|---|---|---|---|---|
| 12-qubit Ising | 3.43x | **4.63x** | 2.04x | 3.43x | 2.32x |
| 10-qubit LiH | 2.31x | **3.24x** | 1.82x | 2.42x | 1.16x |
| 10-qubit QML | 3.76x | **4.38x** | 1.86x | 4.11x | 3.46x |

Table 3: Speedup ratios for all DMD methods for various tasks under the condition of non-smooth optimization .

The speedup ratios in Sec. 6.5 for non-smooth optimization by QuACK from our QuACK with all the DMD methods are given in Table 3.

### E.4  Dynamics Prediction in the Quantum Natural Gradient, Overparameterization, and Smooth Optimizatoin Regimes

We show the dynamics prediction of the DMD prediction in our QuACK framework compared with the intrinsic dynamics of the baseline VQE in the cases of the quantum natural gradient, overparameterization, and smooth optimization regimes, as a supplement of the discussion in Sec. 6.2, 6.3, 6.4.

**Quantum natural gradient.**   As a supplement of Sec. 6.2, in Figure 9, we show the prediction matching of the 10-qubit quantum Ising model (RealAmplitidues ansatz with reps $= 1$) with transverse field $h = 0.5$ using the quantum natural gradient with learning rate 0.01. the prediction from all the DMD methods in our QuACK framework is accurate, as they almost overlap with the intrinsic dynamics of energy from the baseline VQE ($T_{b,t} = 150$). We use $n_{sim} = 10$, $n_{DMD} = 20$ with $n_{iter} = 5$. We choose $n_{SW} = 6$ for SW-DMD, MLP-SW-DMD, and CNN-DMD. For the neural DMD, to achieve good performance, we use 160k training steps for MLP-DMD, and 80k training steps for MLP-SW-DMD and CNN-DMD.

**Overparameterization Regime.**   In Figure 10, we present examples used in Sec. 6.3. The DMD prediction prediction matches the intrinsic dynamics of VQE, with short training ($n_{sim} = 4$) and long prediction ($n_{DMD} = 1000$). Therefore, a great effect of >200x speedup can be achieved.

**Smooth Optimization Regimes.**   In the smooth optimization regimes we consider, the optimization dynamics is not as linear as the cases of the quantum natural gradient and overparameterization. In

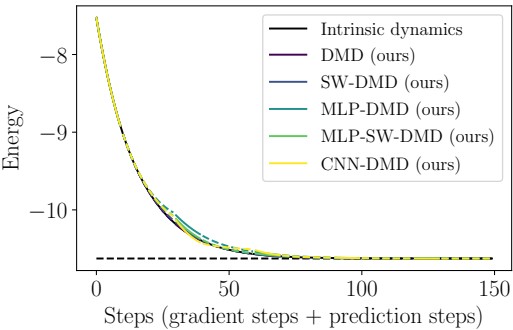

Figure 9: Quantum natural gradient for VQE of 10-qubit quantum Ising model. All the DMD methods using our QuACK framework provide accurate prediction for the intrinsic dynamics of energy of the VQE. For the various DMD methods, solid parts are VQE runs, and the dashed parts are DMD predictions.

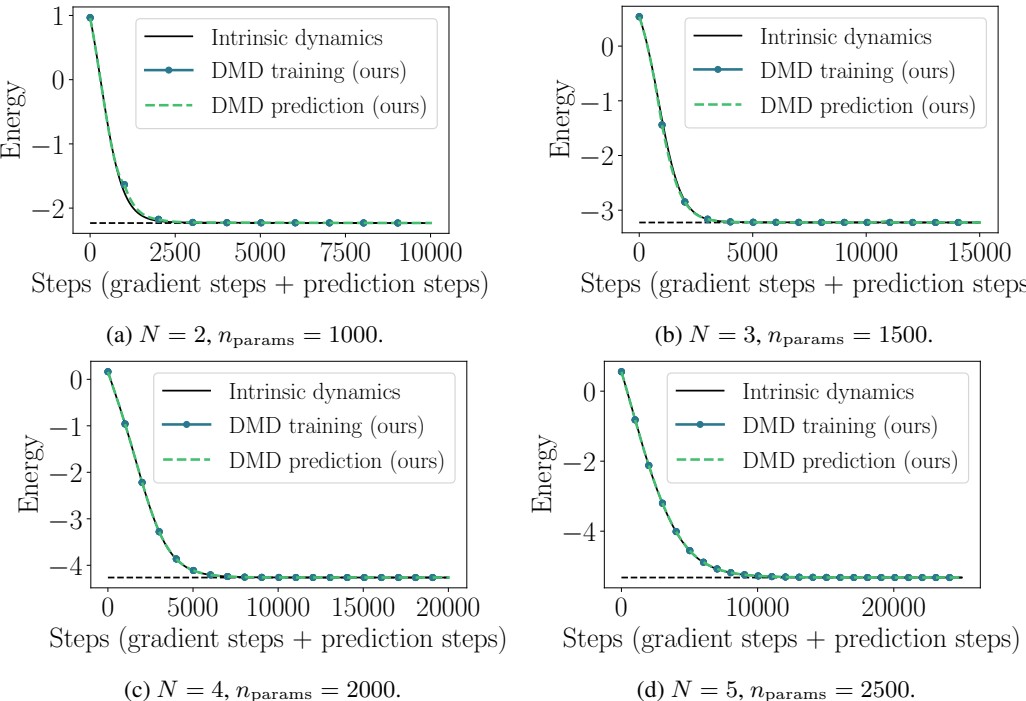

(a) $N = 2$, $n_{\text{params}} = 1000$.

(b) $N = 3$, $n_{\text{params}} = 1500$.

(c) $N = 4$, $n_{\text{params}} = 2000$.

(d) $N = 5$, $n_{\text{params}} = 2500$.

Figure 10: Prediction from DMD in our QuACK framework matches the intrinsic dynamics of energy in the overparameterization regime for VQE of the $N$-qubit quantum Ising model.

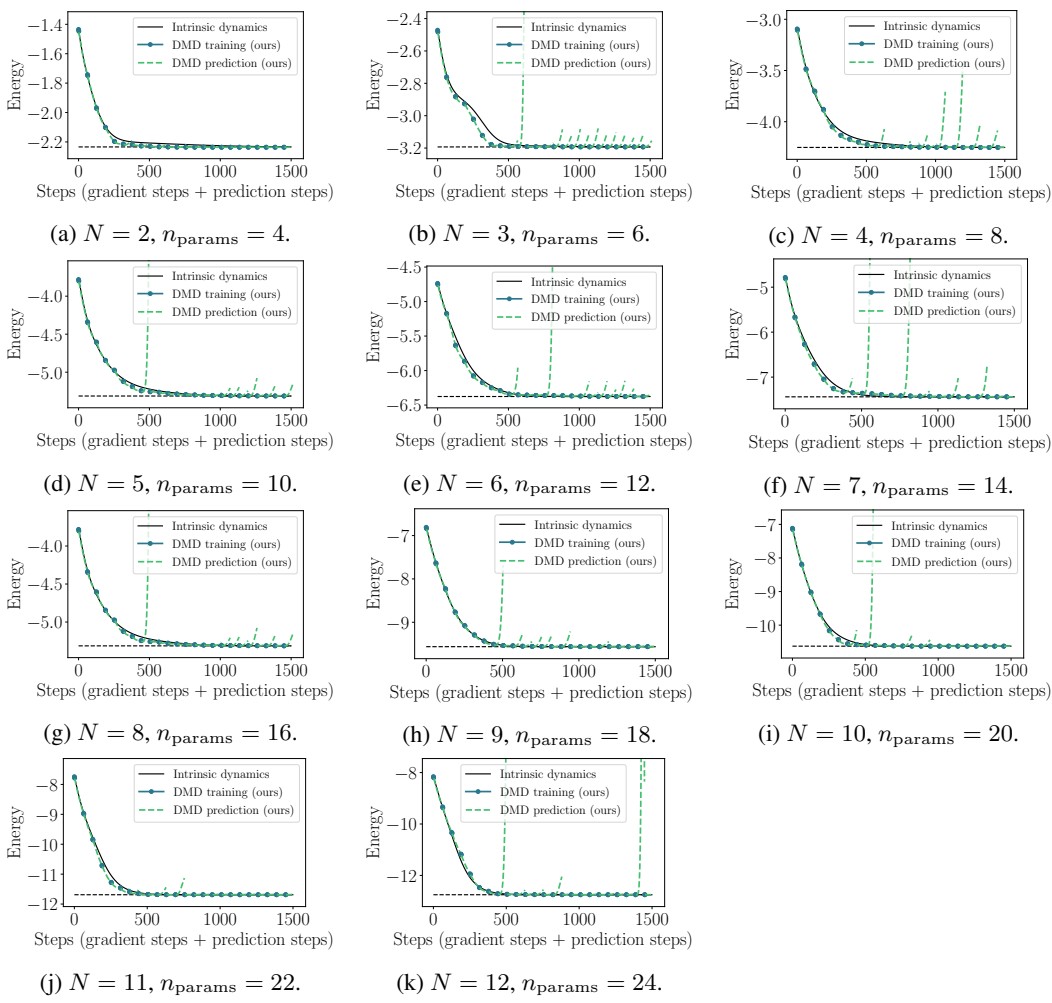

Figure 11: Prediction from DMD in our QuACK framework compared to the intrinsic dynamics of energy in the smooth optimization regime for VQE of the $N$-qubit quantum Ising model with $n_{\text{params}} = 2N$.

addition, the dynamics from the Adam optimizer tends to be more complicated than the quantum natural gradient optimizer and the gradient descent. The above factors make challenges for the DMD prediction. In Figure 11, we present examples used in Sec. 6.4. The DMD prediction, with short training ($n_{\text{sim}} = 3$) and long prediction ($n_{\text{DMD}} = 60$), still captures the trend of decrease of the loss (energy) and has a good alignment in most cases. In some cases when the loss is almost already converged at a late optimization time with steps $\gtrsim 500$, the rate of change of $\boldsymbol{\theta}$ is slow so that the DMD might overfit the flat dynamics and have divergence in its prediction. However, this is not a problem for the practical thanks to the robustness of the alternating controlled scheme in our QuACK, manifestly explained in the example of Figure 6b. Therefore, a great acceleration with >10x speedup can still be achieved. We also comment that intriguingly, in some examples of Figure 11, *e.g.*, 11b , the loss from DMD in our QuACK might decrease faster than baseline VQE, which could lead to $a > 1$ and a higher speedup, as we have mentioned in Sec. 5.2.

## E.5 Experiments with Noise

In Sec. 6.6 and this section, we use the RealAmplitudes ansatz with reps $= 1$ ($n_{\text{params}} = 2N$). The task is VQE of the quantum Ising model with $h = 0.5$ initialized at a random initialization (with a fixed random seed for comparing different cases with the same $N$). The optimizer is Adam with the

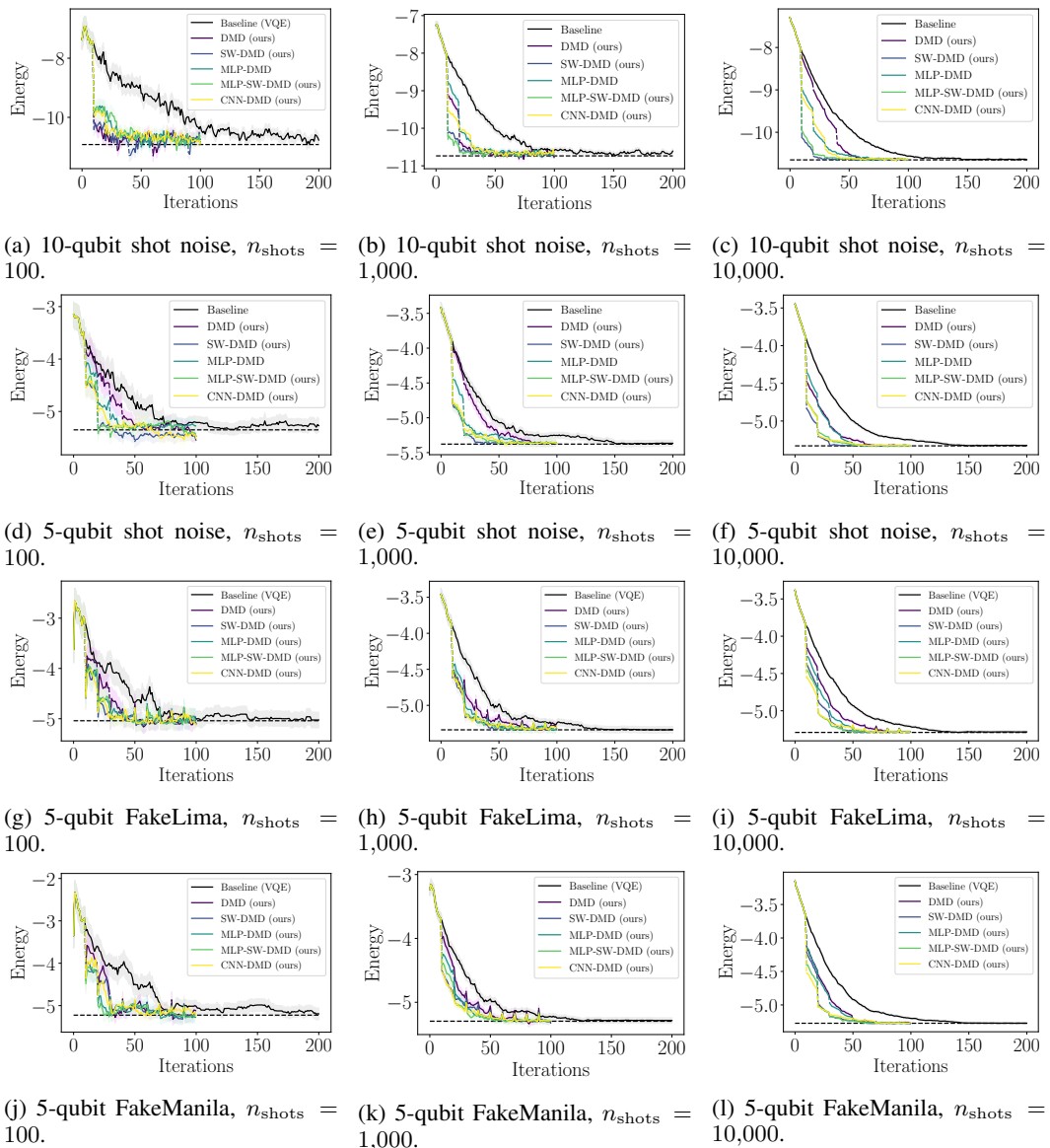

(a) 10-qubit shot noise, $n_{\mathrm{shots}} = 100$.

(b) 10-qubit shot noise, $n_{\mathrm{shots}} = 1,000$.

(c) 10-qubit shot noise, $n_{\mathrm{shots}} = 10,000$.

(d) 5-qubit shot noise, $n_{\mathrm{shots}} = 100$.

(e) 5-qubit shot noise, $n_{\mathrm{shots}} = 1,000$.

(f) 5-qubit shot noise, $n_{\mathrm{shots}} = 10,000$.

(g) 5-qubit FakeLima, $n_{\mathrm{shots}} = 100$.

(h) 5-qubit FakeLima, $n_{\mathrm{shots}} = 1,000$.

(i) 5-qubit FakeLima, $n_{\mathrm{shots}} = 10,000$.

(j) 5-qubit FakeManila, $n_{\mathrm{shots}} = 100$.

(k) 5-qubit FakeManila, $n_{\mathrm{shots}} = 1,000$.

(l) 5-qubit FakeManila, $n_{\mathrm{shots}} = 10,000$.

Figure 12: Experiments with different noise systems at $n_{\mathrm{shots}} = 100, 1000, 10000$. Optimization histories for energy of full VQE and various DMD methods with our QuACK are shown. Dashed lines indicate that the DMD methods are used to accelerate the optimization. The error bands are the statistical uncertainty of from finite-$n_{\mathrm{shots}}$ quantum measurements.

learning rate 0.01. We have also applied measurement error mitigation [5] to reduce the effect of quantum noise.

In Figure 12 with statistical error from shots plotted as error bands, we show the experiments with the acceleration effects by our QuACK framework using all the DMD methods with different noise systems (10-qubit and 5-qubit shots noise, FakeLima, and FakeManila) at $n_{\mathrm{shots}} = 100, 1000, 10000$. The full baseline VQE has $T_{b,t} = 200$ iterations. We choose $n_{\mathrm{sim}} = 10$ and $n_{\mathrm{DMD}} = 20$. SW-DMD, MLP-SW-DMD, and CNN-DMD use window size $n_{\mathrm{SW}} = 6$. The acceleration effects with numerical speedups are given in Table 1 of main text. There are more fluctuations in the small $n_{\mathrm{shots}}$ cases, but our QuACK is still able to accelerate the VQE.

The spikes in the DMD results occur every $n_{\mathrm{sim}} = 10$ iterations at the beginning of every piece of VQE, and are more distinct when $n_{\mathrm{shots}}$ is smaller. This may be because it is harder to make

predictions of the VQE history when the parameter updates time series from measurement is more noisy. Despite the occasional spikes in energy, our QuACK with all the DMD methods still helps to get closer to the better parameters for optimization in later steps.

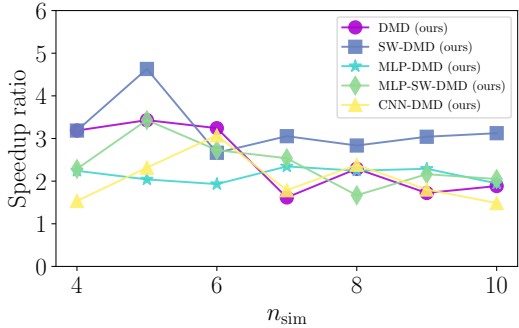

Figure 13: Ablations of $n_{\text{sim}}$ on the 12-qubit Ising model with Adam for all DMD methods.

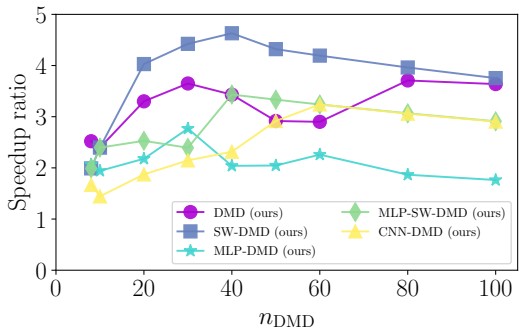

Figure 14: Ablations of $n_{\text{DMD}}$ on the 12-qubit Ising model with Adam for all DMD methods.

### E.6 Ablation Study for $n_{\text{sim}}$ and $n_{\text{DMD}}$

In Figures 13 and 14, we show the ablation study of $n_{\text{sim}}$ and $n_{\text{DMD}}$ of all the DMD methods with our QuACK framework. The experimental setup is the same as the experiment in Sec. 6.5, 12-qubit Ising model with Adam (including having the same initial point), except for varying $n_{\text{sim}}$. (For CNN-DMD, we increase $n_{\text{iter}}$ from 12 to 20 to ensure convergence.) With $n_{\text{sim}} \in [4, 10]$ in Figure 13 and $n_{\text{DMD}} \in [8, 100]$ in Figure 14, all the DMD methods have acceleration effects (>1x speedup), which shows that our QuACK is robust in a range of $n_{\text{sim}}$. In Sec. 6.5 for the quantum Ising model, we have used $n_{\text{sim}} = 5$ and $n_{\text{DMD}} = 40$. The choice of $n_{\text{sim}}$ and $n_{\text{DMD}}$ will affect the speedup, and understanding the behavior from varying $n_{\text{sim}}$ and $n_{\text{DMD}}$ needs future study.

### E.7 Experiments on the Real IBM Quantum Computer Lima

In the current era, the practical cost of using the real quantum resources is still very high (both in waiting time and actual expense), which makes implementing gradient-based optimizers on real quantum hardware an expensive task. However, we still try our best to perform an experiment on real IBM quantum computer Lima and demonstrate how our QuACK can accelerate optimization in complicated settings.

Given our limited access to real quantum hardware, gradient-based optimization with quantum computer is too costly and time consuming. We use the gradient-free optimizer SPSA as an alternative, which updates the parameters by random perturbation. Before we dive in the data, We would like to point out that the gradient-free optimizer is very different from gradient-based optimizer and our main theoretical advantages in this work are for gradient-based quantum optimization. In addition, we find that the real quantum hardware is also subject to fluctuation of day-to-day calibration which could affect the performance of our approach (it can be removed if one has direct timely access to

the quantum computer). Hence, the results in this section should not be directly compared to the experiments in our main text.

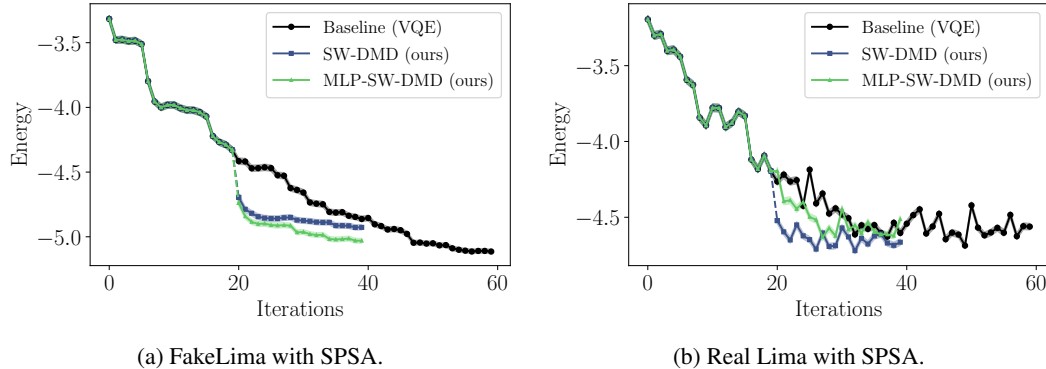

(a) FakeLima with SPSA.
(b) Real Lima with SPSA.

Figure 15: For the 5-qubit quantum Ising model at $h = 0.5$, SPSA on (a) FakeLima and (b) the real quantum computer IBM Lima with $n_{\text{shots}} = 10,000$. Optimization histories for energy of full VQE, SW-DMD, and MLP-SW-DMD are shown. Dashed lines indicate that the DMD methods are used to accelerate the optimization. The error bands are the statistical uncertainty of from finite-$n_{\text{shots}}$ quantum measurements.

We use $n_{\text{shots}} = 10,000$ for the 5-qubit Ising model at $h = 0.5$ on both FakeLima and real Lima. For our QuACK, we use SW-DMD and MLP-SW-DMD with $n_{\text{SW}} = 15$, $n_{\text{sim}} = 20$, $n_{\text{DMD}} = 20$ due to our limited access to the real quantum hardware. We show the results in Figure 15b. Moreover, both the baseline VQE and our QuACK have more fluctuation on real Lima than FakeLima, which can be due to experimental instability in the real lab apparatus, such as day-to-day calibration. However, applying QuACK to this real physical case still help accelerate VQE, as SW-DMD and MLP-SW-DMD partly predict the dynamics and decrease the loss faster than the baseline. This is a positive message as our approach can also be helpful for gradient-free optimization which our theorems are not designed for. Meanwhile, we would like to point out this is not the main focus of this paper because the gradient-based quantum optimization has been shown to have convergence guarantee and better than gradient-free methods [37, 44, 87, 38]. In the coming few years when the quantum hardware become more suitable for gradient-based optimization, our QuACK will demonstrate more advantage for practical applications.

## F  Additional Information

### F.1  Ethics Statement

This work introduces the Quantum-circuit Alternating Controlled Koopman learning (QuACK), a novel algorithm enhancing the efficiency of quantum optimization and machine learning. As a significant advancement in quantum computing, QuACK promises benefits across numerous scientific and technological applications. However, the potential misuse of such technology, particularly in areas like quantum chemistry, could lead to unintended consequences. For example, the increased efficiency in quantum chemistry simulations could inadvertently facilitate the development of chemical weapons if used irresponsibly.

### F.2  Compute

All of our experiments for classically simulating quantum computation are performed on a single CPU. These includes all the experiments presented in the main text and the Appendix, except for the experiment run on the IBM quantum computer *ibmq_lima*, which is one of the IBM Quantum Falcon Processors. The views expressed are those of the authors, and do not reflect the official policy or position of IBM or the IBM Quantum team.

### F.3 Reproducibility

We have stated all of our hyperparameter choices for the experimental settings in the main text and in the appendix. We perform simulations of VQE using Qiskit [2], a python framework for quantum computation, and Yao [47], a framework for quantum algorithms in Julia [7]. Our neural network code is based on Qiskit and Pytorch [57]. Our implementation of quantum machine learning is based on Julia Yao. We use the quantum chemistry module from Pennylane [6] to obtain the Hamiltonian of the molecule. Our code is available at `https://github.com/qkoopman/QuACK`.

### F.4 Broader Impact

The development and implementation of the our QuACK framework could be impactful to the broader scientific and technological community. This approach to quantum optimization can drastically accelerate a multitude of computations in quantum chemistry, quantum condensed matter, and quantum machine learning, which can lead to groundbreaking advancements in these fields.

For instance, in quantum chemistry, the rapid optimization enabled by QuACK could expedite the design of new molecules and materials, potentially leading to the creation of novel drugs, eco-friendly fuels, and advanced materials with superior properties. In quantum condensed matter, faster computations could expedite our understanding of complex quantum systems, possibly catalyzing new insights in physics and material science.

In the realm of quantum machine learning, this acceleration could potentially revolutionize our ability to deal with large datasets and complex models, ultimately leading to more accurate and efficient machine learning algorithms. This, in turn, could benefit a broad range of sectors, from healthcare, where it could be used to better diagnose and treat diseases, to finance, where it could be used to model and predict market trends.

Moreover, the increased speed of quantum optimization could result in energy savings and reduced computing time, contributing to a more sustainable and efficient use of computational resources.

