# OpenReview forum: "QuACK: Accelerating Gradient-Based Quantum Optimization with Koopman Operator Learning"
_NeurIPS.cc/2023/Conference — NeurIPS 2023 spotlight_

### Official Review · Reviewer_ePBi · 2023-06-21

**Soundness:** 3 good
**Presentation:** 3 good
**Contribution:** 3 good
**Rating:** 7
**Confidence:** 4

**Summary:**

This paper utilizes Koopman operator theory to accelerate the optimization of quantum circuits. Stability analysis is performed and the benefit of alternating between standard and Koopman optimization is discussed. A number of experiments are performed, across several different areas of interest to the quantum computing community, with Koopman optimization performing significantly faster than standard approaches. The effect of noise is studied, with QuACK again finding a benefit.

**Strengths:**

1. This paper motivates the need for efficient, gradient based methods for optimizing quantum circuits.

2. This paper shows, via a number of experiments, the potential of Koopman operator theory to accelerate training, over a number of different regimes.

3. This paper tests multiple implementations for computing the Koopman spectra, including DMD, sliding DMD, and machine learning implementations.

4. The Appendix contains many details on quantum computing and the architectures used (the availability of this information should be mentioned in the main text).

**Weaknesses:**

1. While Theorem 5.1 is true, in the sense that any finite numerical implementation of Koopman mode decomposition will find eigenvalues that are not exactly equal to 1, that does not mean that the eigenvalues computed by DMD will not be very close to 1 (see Redman et al., 2021 Fig. 4 for example of eigenvalue being 1.000), and that the associated Koopman mode could be a good approximation to a true fixed point.

2. Similarly, Theorem 5.2 and Corollary 5.4, while true, does not imply that QuACK will not lead the quantum circuit to converge to a different local minima than the gradient-based optimizer. This should be noted explicitly.

3.  Fig. 3 shows that the Koopman based methods converge very quickly to the asymptotic energy and test accuracy - from just the plots alone it seems like the speed-up should be greater than the reported 4x.  Why is this the case?

4. The Koopman learning section should be expanded to include mention of the Koopman mode decomposition (Mezic, 2005). Additionally, the need for data to computed the Koopman mode decomposition should be motivated in Sec. 3, before the QuACK algorithm (which needs data from the gradient based optimizer) is presented. The interested reader should be referred to more detailed discussion in Budisic  et al., 2012 and Brunton et al., 2022. Finally, it should be noted that the idea of time-delays is related to Taken's theorem, with the implementation presented in Eq. 1 being Hankel DMD (Arbabi and Mezic, 2017). The sliding window DMD method is similar to streaming DMD (Hermati et al., 2014; Giannakis et al., 2023).

5. While this is, to my knowledge, the first approach of using Koopman operator theory to the study of quantum machine learning, it would be appropriate to cite more of the work on studying machine learning using Koopman operator theory (e.g., Mohr et al., 2020; Naiman and Azencot, 2021; Šimánek et al., 2022; Liang et al. 2022). Additionally, the previous work that has optimized neural networks using Koopman operator theory (Dogra and Redman, 2020; Tano et al., 2020) should be discussed in more detail (especially when presenting the speed-up $s$, which was defined similarly by Dogra and Redman, 2020, and when discussing the alternating strategy of Koopman training, which Tano et al. 2020 also utilized). Work using Koopman to study iterative algorithms (Dietrich et al., 2020; Redman et al., 2022) more generally should also be cited when mentioning that "the optimization trajectory...is a dynamical system" (line 52). Relatedly, the references with respect to Koopman operator theory need to be updated. Extended-DMD (EDMD) refers to a specific algorithm, developed by Williams et al. 2015. This was not referenced when EDMD was mentioned in the paper (line 80), and all other papers cited described extensions to DMD that are not EDMD. The connection between DMD and Koopman mode decomposition should be noted (Rowley et al., 2009).

6. Is it true that classical backpropagation is independent of $n_{params}$? I think you mean that you don't need to do individual perturbations along the dimension of each parameter, but this needs to be clarified.

MINOR COMMENTS:

1. The speedup ratio of Fig. 2b and c seem to increase with the number of parameters. Is this expected?

2. Describing Koopman as "renowned for accelerating/predicting nonlinear dynamical systems" (lines 4 and 54) is a little overgeneral. Perhaps, saying "which has found utility in applications, as it allows for a linear representation of nonlinear dynamical systems," would be more specific and accurate.

3. It is mentioned that the quantum natural gradient is linked to imaginary time in the Introduction, but this is not discussed in overview section of Quantum Natural Gradient in Sec. 3.

4. There are a few minor typos:
    i. The question "How can we accelerate...", which is not a yes/no question, is posed (line 48), to which you answer "in the affirmative".
    ii. You list 3 speedups, but four conditions (lines 61, 62).
    iii. "of the system is called wave function" (line 88)
    iv. "the a layerwise partitioning" (line 299). Does this refer to the layer Koopman partitioning of Dogra and Redman, 2020?


**Questions:**

1. Why does the 4x speedup reported for Fig. 3 not match the qualitative increase in speed-up from the figure?

2. Is it true that classical backpropagation is independent of $n_{params}$?

3.  The speedup ratio of Fig. 2b and c seem to increase with the number of parameters. Is this expected?

4. Does "the a layerwise partitioning" (line 299) refer to the layer Koopman partitioning of Dogra and Redman, 2020?


**Limitations:**

The authors do a sufficient job addressing their limitations in the Limitations subsection of Sec. 4. They also study several different examples, across multiple different subregions of quantum computing, as well as different noise condition, providing strong evidence for their claims that Koopman training accelerates quantum optimization.

---

> ### Author Rebuttal · Authors · 2023-08-10
>
> This paper shows, via a number of experiments, ...
>
> The Appendix contains many details on quantum computing and the architectures used (the availability of this information should be mentioned in the main text).
>
> **We thank the reviewer’s appreciation of our contributions.**
>
> Weaknesses:
> While Theorem 5.1 is true, ...
>
> **Thanks for the comment. It is a good point that for certain cases the Koopman mode could be a good approximation to a true fixed point, while for quantum optimization we notice that these could be more subtle. Many hardware-efficient quantum circuits do not have nice structures compared to certain well-designed neural networks like ResNet, and the parameters of quantum circuits are angles which could rotate periodically. In those cases, we find that the asymptotic DMD prediction is more likely to be trivial or unstable, which is the motivation of Theorem 5.1. We have included further discussion in the updated paper to elaborate this point.**
>
> Similarly, Theorem 5.2 and Corollary 5.4, while true, ...
>
> **Thanks for the comment. We agree with the reviewer that for Theorem 5.2 it is possible for QuACK to converge to different local minima compared to gradient-based methods. For the convex problem setup in Corollary 5.4, there will be one global minima so that QuACK and gradient-based optimizer should converge to the same point. We have included this note in the updated paper.**
>
> Fig. 3 shows that the Koopman based methods converge very quickly to the asymptotic energy and test accuracy - from just the plots alone it seems like the speed-up should be greater than the reported 4x. Why is this the case?
>
> **Thanks for the comment. Indeed, this is a plotting issue that appears to converge very quickly, which should be around 4x if we zoom in clearly. We include a zoomed-in plot in Fig. 2 of our one-page pdf.**
>
> The Koopman learning section should be expanded to include ...
>
> **Thanks for the suggestions. We have improved the writing based on the suggestions and included the relevant references.**
>
> While this is, to my knowledge, the first approach of using Koopman operator theory to the study of quantum machine learning...
>
> **Thank you for suggestion these foundational works. We updated our manuscript as follows:**
>
> **- After the “Koopman Theory” paragraph in the “Related Work” Section, we continued with: Ours and the above contributions are built upon works on machine learning using Koopman operator theory\citep{mohr2020koopman,mohr2020predicting,naiman2021koopman,vsimanek2022learning,liang2022credit}.**
>
> **- We added “Similar speed-up definition has been considered in\citep{dogra2020optimizing}” as a footnote in line 256.**
>
> **- We added “The algorithm iteratively alternates the simulation steps with the Koopman steps for $n_\mathrm{iter}$ steps, similarly to\citep{tano2020accelerating}” in line 156.**
>
> **- We added “similarly to\citep{dietrich2020koopman,redman2022algorithmic},” in line 52.**
>
> **- We added “(connected to Koopman mode decomposition\citep{rowley2009spectral})” in line 79.**
>
> Is it true that classical backpropagation is independent of...
>
> **Thanks for the suggestion. The memory complexity of classical backpropagation still scales with the number of parameters but the computation complexity does not, which is the same as the forward propagation. It is correct because we do not need to do individual perturbations along the dimension of each parameter, thanks to automatic differentiation. We have made this clarification in the updated paper.**
>
> MINOR COMMENTS:
>
> The speedup ratio of Fig. 2b and c seem to increase with the number of parameters. Is this expected?
>
> **Thanks for the question. This is expected according to Theorem 5.5. The upper bound in Eq. (3) for the speedup increases as the number of parameters $p$ increases, since $f(p)$ increases when $p$ increases.**
>
> Describing Koopman as "renowned for accelerating/predicting nonlinear dynamical systems" (lines 4 and 54) is ...
>
> **Thanks for the suggestion. We have written the sentences based on the suggestion.**
>
> It is mentioned that the quantum natural gradient...
>
> **Thanks for the suggestion. We have mentioned the connection in the beginning of Section 5 and in the updated version, we have added the discussion on the imaginary time evolution to the Quantum Natural Gradient in the overview section as well.**
>
> There are a few minor typos: i. The question "How can we accelerate...", ...
>
> **Thanks for the suggestions. We have fixed the typos. The layer Koopman partition is inspired by Dogra and Redman, 2020, and here we partition the layers of parameters in the quantum circuit instead of the classical neural networks. We have corrected the typo and included more description of the layer Koopman partition in the updated paper.**
>
> Questions:...
>
> **Thanks for the questions. We have addressed them above.**

---

> > ### Comment · Reviewer_ePBi · 2023-08-11
> > **Response to reviewers**
> >
> > Thank you for your detailed rebuttal and for including items discussed in the review into the paper.
> >
> > I believe the paper has been strengthened and I feel that my questions/concerns have been sufficiently clarified.
> >
> > Based on this and reading all reviews from other reviewers, as well as the authors' responses to these reviewers,  I feel confident in the quality of this paper. I will increase my score.

---

> > > ### Author Response · Authors · 2023-08-14
> > > **Response to Reviewer ePBi**
> > >
> > > We appreciate Reviewer ePBi for reading all the reviews and our responses. We would also like to thank the Reviewer for acknowledging the quality of our paper and increasing our score. If there is any follow-up comment or suggestion, please let us know and we would be glad to discuss and elaborate more.

---

### Official Review · Reviewer_sYsu · 2023-06-22

**Soundness:** 3 good
**Presentation:** 3 good
**Contribution:** 3 good
**Rating:** 6
**Confidence:** 3

**Summary:**

This paper proposes QuACK, which leverages Koopman operator learning theory to accelerate gradient-based quantum optimization and machine learning. QuACK utilizes the sliding window DMD to enrich the embedding, and neural networks to learn and predict the nonlinear dynamics. Experiments on quantum Ising model, quantum chemistry and quantum machine learning show QuACK’s remarkable speedup and robustness. Also, theoretical analysis of its stability, convergence and complexity is provided.

**Strengths:**

1. The proposed QuACK can significantly accelerate gradient-based quantum optimization by predicting the gradient dynamics, which may help deploy gradient-based quantum algorithms on NISQ devices.
2. Extensive experiments on quantum Ising model, quantum chemistry and quantum machine learning are conducted. Besides, speedup is verified for quantum natural gradients, in overparameterized scenes, and on IBM quantum devices.
3. Theoretical analysis of QuACK’s stability, convergence, complexity and speedup is provided.

**Weaknesses:**

1. $\Theta(t_{d+1})\approx K\Phi(\Theta(t_0))$ in page 5 seems to lack theoretical support, since the koopman operator is $Kg(x(t))=g(x(t+1))$, which is different from the given formula.
2. Though gradient-based optimization with quantum computers can be costly and time consuming, an extra experiment on NISQ devices for gradient-based optimization is necessary for validating the feasibility of QuACK on realistic scenes.

**Questions:**

1. The parameter $n_{sim}$ seems not well-defined in the paper. What is the specific meaning of it in Algorithm 1?
2. Have you compared QuACK with gradient-free methods like COBYLA?

**Limitations:**

Yes.

---

> ### Author Rebuttal · Authors · 2023-08-10
>
> Weaknesses:
> Θ(td+1)≈KΦ(Θ(t0)) in page 5 seems to lack theoretical support, since the koopman operator is Kg(x(t))=g(x(t+1), which is different from the given formula.
>
> **Thanks for the question. This is a good point to note. As the reviewer points out, this is different than the conventional Koopman operator notation. Indeed, here we take the perspective of ‘generalized operator’ by viewing K.𝜱 as one operator that transfers the state at time t to the state at time t+1. This formulation also generalizes the standard DMD where 𝜱 is the identity map while it can still be solved efficiently with 𝜱, particularly with the SW-DMD case we can still perform the SVD directly.**
>
> Though gradient-based optimization with quantum computers can be costly and time consuming, an extra experiment on NISQ devices for gradient-based optimization is necessary for validating the feasibility of QuACK on realistic scenes.
>
> **Thanks for the comment. Indeed, the current experiment on NISQ devices using gradient-based optimization is very costly. To provide a concrete example of quantum cost, the quantum computing service of Rigetti on Amazon (https://aws.amazon.com/braket/pricing/) for one VQE quantum optimization experiment of 160 params over 200 steps with 100 shots each can take: \$ 160 * 200 * (0.3 + 100 * 0.00035) = \$ 10720. While we are also very interested in performing gradient-based optimization on NISQ device, which will further demonstrate the benefit of our work on saving both computation and economic resources, we are not able to perform such experiments given our current budget and time and it could be an important direction for future work.**
>
> Questions:
> 1. The parameter n_sim seems not well-defined in the paper. What is the specific meaning of it in Algorithm 1?
>
> **We thank the reviewer for pointing out the unclearness of the definition of $n_\mathrm{sim}$. $n_\mathrm{sim} = m$ in Algorithm 1. We have now used “$n_\mathrm{sim}$” to replace “$m$” in Algorithm 1 and the text near it.**
>
> 2. Have you compared QuACK with gradient-free methods like COBYLA?
>
> **We thank the reviewer for raising this interesting point regarding gradient-free methods. While gradient-based methods and gradient-free methods are applicable types of optimizers for VQAs, gradient-based methods use gradient information and thus can know better about the local geometry [1], especially with quantum Fisher information through quantum natural gradient. Particularly, the gradient method has been shown to converge for quantum optimization [2, 3]. It is also known that other gradient-free methods such as SPSA do not perform well compared to gradient-based methods [4]. Given the nice convergent property of gradient-based optimization, it is meaningful to accelerate gradient-based methods. Hence, our paper focuses on improving upon gradient-based methods in variational quantum algorithms, instead of comparing with gradient-free methods.**
>
> **In Fig. 3 of our one-page pdf, we have included an example of shot-noise of the quantum Ising model to provide a concrete demonstration of the above point. While the gradient-based VQE with Adam finds a lower energy than COBYLA, as COBYLA gets trapped. Our methods further improve upon Adam.**
>
> [1] Li, Hao, Zheng Xu, Gavin Taylor, Christoph Studer, and Tom Goldstein. "Visualizing the loss landscape of neural nets." Advances in neural information processing systems 31 (2018).
> [2] Ryan Sweke, Frederik Wilde, Johannes Meyer, Maria Schuld, Paul K. Faehrmann, Barthélémy Meynard-Piganeau, Jens Eisert. Stochastic gradient descent for hybrid quantum-classical optimization. Quantum 4, 314 (2020).
> [3] You, Xuchen, Shouvanik Chakrabarti, and Xiaodi Wu. "A convergence theory for over-parameterized variational quantum eigensolvers." arXiv preprint arXiv:2205.12481 (2022).
> [4] Leng, Jiaqi, Yuxiang Peng, Yi-Ling Qiao, Ming Lin, and Xiaodi Wu. "Differentiable analog quantum computing for optimization and control." Advances in Neural Information Processing Systems 35 (2022): 4707-4721.

---

> > ### Comment · Reviewer_sYsu · 2023-08-17
> >
> > Thanks for addressing most of my concerns. Based on the other reviewers' opinions and your rebuttals, I will raise my score.

---

> > > ### Author Response · Authors · 2023-08-21
> > > **Response to Reviewer sYsu**
> > >
> > > We would like to thank again Reviewer sYsu for helpful comments on improving our work and acknowledging our rebuttal and raising our score.

---

### Official Review · Reviewer_kkUu · 2023-07-03

**Soundness:** 3 good
**Presentation:** 3 good
**Contribution:** 3 good
**Rating:** 6
**Confidence:** 5

**Summary:**

The paper presents an innovative framework known as Quantum-circuit Alternating Controlled Koopman learning (QuACK), which effectively combines Koopman operator theory and natural gradient methods in quantum optimization. Through the utilization of an alternating algorithm, QuACK allows for the efficient prediction of gradient dynamics on quantum computers, resulting in a notable acceleration of variational quantum algorithms (VQAs). The authors provide empirical evidence showcasing the effectiveness of QuACK across diverse VQA tasks, including those involving noisy environments.






**Strengths:**

The paper's strength lies in its contribution to accelerating the optimization of variational quantum algorithms (VQAs), which is crucial for leveraging the capabilities of modern quantum machines in practical problem-solving. The proposed QuACK addresses this challenge and presents intriguing findings. By incorporating Koopman operator theory and deep neural networks, QuACK offers noticeable speed improvements compared to conventional optimization methods like the parameter shift rule. The numerical results provided in the paper demonstrate the efficacy of QuACK.

**Weaknesses:**

The primary concern of the paper lies in the stability and robustness of QuACK. Algorithm 1 suggests that the performance of QuACK may heavily rely on the hyperparameters n_sim and n_DMD. Determining the appropriate values for these hyperparameters is an empirical process and may require additional hyperparameter optimization, introducing a potential computational burden. While the authors have conducted extensive numerical analysis, it remains unclear whether an improper setting of these hyperparameters could undermine the advancements of QuACK in optimizing modest-sized VQAs that aim to achieve quantum advantages, particularly in scenarios involving 100+ qubits.

Furthermore, the issue of stability is exacerbated by QuACK's sensitivity to the system and shot noise. As demonstrated in the study by Du et al. (PRX Quantum 2.4, 2021: 040337), gradient-based methods can exhibit divergence under significant noise and low shot numbers. It remains uncertain whether QuACK demonstrates greater robustness compared to standard optimization strategies in such challenging scenarios. Addressing these stability and robustness concerns would augment the applicability and reliability of QuACK in practical quantum optimization tasks. A more in-depth discussion of these issues would be preferable.

In addition, there are some minor issues with the paper:

- The font size in Figure 1 is too small, making it challenging to read.
- The relation between n_sim and m is not clearly defined in the paper, and clarification is needed.
- Corollary 5.4 could be removed since it is redundant given that the optimization of VQAs is generally non-convex.
- The format of references does not adhere to the standard style in NeurIPS.
- Several references are missing, including:
   1. You, Xuchen, et al. arXiv:2303.14844.
   2. García-Martín, D., Larocca, M., & Cerezo, M. arXiv:2302.05059.
   3. Wang, Xinbiao, et al. arXiv:2208.14057.

**Questions:**

The concerns listed in *Weakness* should be adequately addressed. Satisfactory responses to these points will further enhance the quality of the paper.

**Limitations:**

The authors partially addressed the limitations of QuACK.

---

> ### Author Rebuttal · Authors · 2023-08-10
>
> Strengths:
> The paper's strength lies in its contribution to accelerating the optimization of variational quantum algorithms (VQAs), which is crucial for leveraging the capabilities of modern quantum machines in practical problem-solving. The proposed QuACK addresses this challenge and presents intriguing findings. By incorporating Koopman operator theory and deep neural networks, QuACK offers noticeable speed improvements compared to conventional optimization methods like the parameter shift rule. The numerical results provided in the paper demonstrate the efficacy of QuACK.
>
> **We thank the Reviewer’s appreciation of our contributions.**
>
> Weaknesses:
> The primary concern of the paper lies in the stability and robustness of QuACK. Algorithm 1 suggests that the performance of QuACK may heavily rely on the hyperparameters n_sim and n_DMD. Determining the appropriate values for these hyperparameters is an empirical process and may require additional hyperparameter optimization, introducing a potential computational burden. While the authors have conducted extensive numerical analysis, it remains unclear whether an improper setting of these hyperparameters could undermine the advancements of QuACK in optimizing modest-sized VQAs that aim to achieve quantum advantages, particularly in scenarios involving 100+ qubits.
>
> **We thank the Reviewer for the question. While n_sim and n_DMD are hyperparameters, we have certain intuition and rules of thumb for choice. In general, for smooth dynamics (such as the overparameterization regime), we need smaller n_sim, while for more complicated dynamics, larger n_sim is needed to capture the non-smooth dynamics. To gain a better understanding of the effects of n_sim, we have already performed an ablation study on n_sim in our supplementary materials. We have further performed an ablation study for n_DMD in Fig. 1 of our one-page pdf. The results show that our QuACK are robust to the change in n_sim and n_DMD, consistent with our intuition. For example, in the 12-qubit Ising model, n_DMD between 20-100 achieve good speedup. Meanwhile, as we mention in the Limitation part of our paper, more theoretical and algorithmic study of the hyperparameter choice will be an important future direction. For example, one can consider the online learning techniques in ML to adaptively update the hyperparameters for better performance. We have included the new results and discussions in the updated paper.**
>
> Furthermore, the issue of stability is exacerbated by QuACK's sensitivity to the system and shot noise. As demonstrated in the study by Du et al. (PRX Quantum 2.4, 2021: 040337), gradient-based methods can exhibit divergence under significant noise and low shot numbers. It remains uncertain whether QuACK demonstrates greater robustness compared to standard optimization strategies in such challenging scenarios. Addressing these stability and robustness concerns would augment the applicability and reliability of QuACK in practical quantum optimization tasks. A more in-depth discussion of these issues would be preferable.
>
> **Thanks for the comment. It is a good point to have understanding on the noise effects. While gradient-based methods may diverge under significant noise and low shot numbers, it is not clear if quantum optimization is useful in such situations since the quantum computers could just be too noisy to use. To understand the robustness of our approach under both quantum noise and shot noise, we have also performed ablation study in Section 6.6 with Table 1 and Figure 4. We find that our approach can still achieve 2x to 5.5x Speedup in different noisy regimes. Indeed, the alternating control scheme we developed is helpful to improve the robustness since it uses the optimal value for the next learning starting point, which reduces the noise fluctuation for a particular step. We have included more discussion on the topic in the updated paper.**
>
> In addition, there are some minor issues with the paper:
>
> The font size in Figure 1 is too small, making it challenging to read.
>
> **Thanks for the suggestion and we have increased the font size.**
>
> The relation between n_sim and m is not clearly defined in the paper, and clarification is needed.
>
> **We thank the reviewer for pointing this out. For that purpose we update the paper in Algorithm 1 by adding the following specification of the hyperparameters: “Koopman operator learning parameters $m$ (num. simulation. steps $n_{\mathrm{sim}})$, $n_{\mathrm{DMD}}$ (num. DMD steps).”**
>
> Corollary 5.4 could be removed since it is redundant given that the optimization of VQAs is generally non-convex.
>
> **Thanks for the suggestion. It is true that VQA is generally non-convex and the original motivation is to show QuACK maintains nice properties for convex optimization as a theoretical result, in case readers are interested in.  We will be glad to follow the suggestion and move Corollary 5.4 to Appendix.**
>
> The format of references does not adhere to the standard style in NeurIPS.
>
> **Thanks for the suggestions. We modified the format of references to enumerate with numbers.**
>
> Several references are missing, including:
> You, Xuchen, et al. arXiv:2303.14844.
> García-Martín, D., Larocca, M., & Cerezo, M. arXiv:2302.05059.
> Wang, Xinbiao, et al. arXiv:2208.14057.
>
> **Thank you for the valuable suggestion for including these references. We included the following paragraph after the “Quantum Optimization Methods” paragraph in the Related Work Section:**
>
> ***Some additional lines of work are related to our study. \citet{you2023analyzing} analyzes the convergence of quantum neural networks through the lens of the neural tangent kernel. Similarly, working through an effective quantum NTK theory, \citet{wang2022symmetric} proposes symmetric pruning to improve the loss landscape and the convergence of quantum neural networks. Finally, \citet{garcia2023effects} studies the effect of noise on overparameterization in quantum neural networks.***

---

> > ### Comment · Reviewer_kkUu · 2023-08-18
> >
> > I thank the authors for their careful response, which adequately addresses all my concerns.

---

> > > ### Author Response · Authors · 2023-08-21
> > > **Response to Reviewer kkUu**
> > >
> > > We would like to thank again Reviewer kkUu for helpful comments on improving our work and acknowledging our rebuttal.

---

### Official Review · Reviewer_M6jg · 2023-07-05

**Soundness:** 4 excellent
**Presentation:** 3 good
**Contribution:** 3 good
**Rating:** 7
**Confidence:** 3

**Summary:**

The paper presents an algorithm that  effectively "extends" $k$ gradients updates to $k'$ gradient updates by fitting a neural koopman operator. The additional cost is made worth it because the algorithm is derived in a quantum machine learning setting where quantum gradients are expensive and limited and classical operations are comparatively free and unlimited. The algorithm successfully speeds up Quantum Natural Gradient, in particular in the quantum overparametrzation regime (200x) but even in nonsmooth and noisy regimed it achieves 2x to 3x speedup. Convergence proofs for convex problems and some other theoretical results augment the derivation and empirical evaluation.

**Strengths:**

Originality: people don't use koopman operators enough, using them as a framework to perform a $m$-step extension of $k$ expensive gradient steps is  very clever and I might steal the idea for some of my own work post review period
Quality: Argument is clear and sufficient analysis is done to sanity check and gain insight into the algorithm, but I would call this an empirical paper still, not in the least due to the training tricks required to make the neural koopman estimate work. For the empiricism, full marks, good ablations in the appendix, error bars where it matters. Bonus points for an attempt on real quantum experiments, even if the experiments are kind of tangential to the method presented (gradient free vs. gradient dynamics estimation)
Clarity:  well written, easy to understand
Significance: speaking as an ML person, I think this is a significant contribution to QML, in particular if we remain constraint by quantum operations




**Weaknesses:**

- what is the wallclock time of each method? this is a relatively crucial detail I am missing for context


**Questions:**

- what is the distinction between previous "neural koopman" works (e.g. lusch et al) and this work? is it "just" that it is applied to QML learning dynamics?

**Limitations:**

I think limitations are well stated

---

> ### Author Rebuttal · Authors · 2023-08-10
>
> Strengths:
>
> Originality: people don't use koopman operators enough, using them as a framework to perform a -step extension of  expensive gradient steps is very clever and I might steal the idea for some of my own work post review period Quality: Argument is clear and sufficient analysis is done to sanity check and gain insight into the algorithm, but I would call this an empirical paper still, not in the least due to the training tricks required to make the neural koopman estimate work. For the empiricism, full marks, good ablations in the appendix, error bars where it matters. Bonus points for an attempt on real quantum experiments, even if the experiments are kind of tangential to the method presented (gradient free vs. gradient dynamics estimation) Clarity: well written, easy to understand Significance: speaking as an ML person, I think this is a significant contribution to QML, in particular if we remain constraint by quantum operations
>
> **We thank the reviewer’s appreciation of our contributions.**
>
> Weaknesses:
> what is the wallclock time of each method? this is a relatively crucial detail I am missing for context
>
> **Thanks for the question. Here we define the wallclock time as the time of DMD learning + the time of quantum optimization. The DMD learning on classical computers is indeed very efficient (for each piece, only milliseconds for DMD and SW-DMD, and less than 1 minute for neural DMD) and thus it is not the bottleneck. The main bottleneck is the time of quantum optimization, and our QuACK is designed to speed up the time of quantum optimization. In our work, the quantum optimization is simulated by classical computers. The simulations on a classical computer (a single CPU) take the time at orders from minutes to an hour depending on the details of the task and hyperparameters. As a representative example, for the 12-qubit Ising model with Adam in Sec. 6.5 Figure 3(b) the time costs of simulations (with the parameter-shift gradient updates explicitly to mimic the actual quantum optimization) are**
> | Method      | Wallclock time |
> | ----------- | ----------- |
> | VQE (baseline)      |   2866 seconds    |
> |  DMD (ours)     | 646 seconds    |
> |  SW-DMD (ours)   |  663 seconds       |
> |  MLP-DMD (ours) | 818 seconds |
> | SW-MLP-DMD (ours) | 849 seconds |
> | CNN-DMD (ours) |  828 seconds   |
>
> **Thanks to the acceleration by our QuACK, we can see the benefit of having less wallclock time by our various DMD methods compared to the baseline VQE. We note that these wallclock times are for classical simulations of quantum optimization. For actual quantum optimization, it is expected that our approach can gain much more benefit, since the dominant cost is from each gradient step of quantum optimization. The wallclock time saved on an actual quantum computer will be proportional to the speedup factor shown in our paper multiplying the actual time of each iteration in quantum optimization. This will help us to save more resources since the actual quantum optimization time is more expensive than the classical simulations of the quantum optimization.**
>
> **We have included these details in the updated paper.**
>
> Questions:
> what is the distinction between previous "neural koopman" works (e.g. lusch et al) and this work? is it "just" that it is applied to QML learning dynamics?
>
> **Thanks for the questions. While our neural DMD methods are inspired by the previous “neural koopman” works (e.g., lusch et al), there are certain features of our work. First, our work focuses on the acceleration of quantum optimization and QML, while the previous one is designed to learn classical PDEs. To achieve our goal, we need to analyze the quantum gradient optimization properties with the DMD approach, which builds connections among quantum imaginary time evolution, overparameterization theory, and complexity analysis. Furthermore, to achieve robust and efficient optimization, we develop an alternating control scheme to make sure our methods are always at least as good as the standard gradient-based method, which does not exist in the previous work.**

---

> > ### Comment · Reviewer_M6jg · 2023-08-18
> >
> > Thank your for addressing my questions.

---

> > > ### Author Response · Authors · 2023-08-21
> > > **Response to Reviewer M6jg**
> > >
> > > We would like to thank again Reviewer M6jg for helpful comments on improving our work and acknowledging our rebuttal.

---

### Author Rebuttal · Authors · 2023-08-10

**We thank the reviewers’ comments and suggestions. Reviewer M6jg acknowledges originality, clarity of our argument, “full marks” on our experiments, experiments on real hardware, and significance of the results. Reviewer kkUu acknowledges “intriguing findings” in approaching the limitations of crucial approaches to modern quantum machines in practical problem-solving, noticeable improvements in our experiments, and the efficacy of our framework. Reviewer sYsu acknowledges significant acceleration of our method, extensive experiments and theoretical analysis. Reviewer ePBi acknowledges the motivation, demonstration of promise, multiple implementations, and extensive ablations of our method.**

**Some questions and suggestions were posed regarding topics, including wallclock times (M6jg), robustness of our method (kkUu), theory alignment with known Koopman theory (sYsu), experiments on real hardware (sYsu), clarifications throughout the text (ePBi).**

**We have addressed all of them with additional experiment and discussion, presented in our individual responses. We easily incorporated all of our responses in our manuscript, and we described how we accomplished that in our individual responses to the reviewers.**

**Thus, we hope that the reviewers can consider raising their scores.**

---

### Decision · Program_Chairs · 2023-09-21

**Decision:**

Accept (spotlight)

**Comment:**

The paper provides a new approach for quantum optimization that permits efficient evaluation of gradients. The approach particularly excels in the overparameterized regime which is particularly of interest for modern deep learning methods. The reviewers uniformly appreciated the originality and empirical effectiveness of the proposed approach, and are optimistic that the approach will inspire many followups in quantum optimization. There were some small concerns regarding hyperparameters and theoretical justifications, which were addressed adequately during the discussion phase. The AC would recommend including these details in the camera-ready version. Overall, there was broad consensus for accepting the paper.